# IMPRESS: Evaluating the Resilience of Imperceptible Perturbations Against Unauthorized Data Usage in Diffusion-Based Generative AI

**Bochuan Cao**
The Pennsylvania State University
bccao@psu.edu

**Changjiang Li**
Stony Brook University
meet.cjli@gmail.com

**Ting Wang**
Stony Brook University
inbox.ting@gmail.com

**Jinyuan Jia**
The Pennsylvania State University
jinyuan@psu.edu

**Bo Li**
University of Illinois, Urbana Champaign
lbo@illinois.edu

**Jinghui Chen**
The Pennsylvania State University
jzc5917@psu.edu

## Abstract

Diffusion-based image generation models, such as Stable Diffusion or DALL·E 2, are able to learn from given images and generate high-quality samples following the guidance from prompts. For instance, they can be used to create artistic images that mimic the style of an artist based on his/her original artworks or to maliciously edit the original images for fake content. However, such ability also brings serious ethical issues without proper authorization from the owner of the original images. In response, several attempts have been made to protect the original images from such unauthorized data usage by adding imperceptible perturbations, which are designed to mislead the diffusion model and make it unable to properly generate new samples. In this work, we introduce a perturbation purification platform, named IMPRESS, to evaluate the effectiveness of imperceptible perturbations as a protective measure. IMPRESS is based on the key observation that imperceptible perturbations could lead to a perceptible inconsistency between the original image and the diffusion-reconstructed image, which can be used to devise a new optimization strategy for purifying the image, which may weaken the protection of the original image from unauthorized data usage (e.g., style mimicking, malicious editing). The proposed IMPRESS platform offers a comprehensive evaluation of several contemporary protection methods, and can be used as an evaluation platform for future protection methods.

## 1 Introduction

Diffusion-based image generation models, e.g., Stable Diffusion (Rombach et al., 2022b) and DALL-E 2 (Ramesh et al., 2022), have gained increasing attention due to their exceptional performance in synthesizing high-quality samples by leveraging given images. Despite their superior performance, existing studies (Shan et al., 2023) showed that they could be abused to generate new images without proper authorization from the original data owner. For instance, they could be used to learn from the original artworks of a specific artist and generate artistic images that mimic his/her style without authorization, which may lead to violation of intellectual property or copyrights (Dixit, 2023; Joseph Saveri Law Firm LLP, 2023; Edwards, 2022). They can also be used to maliciously edit the images

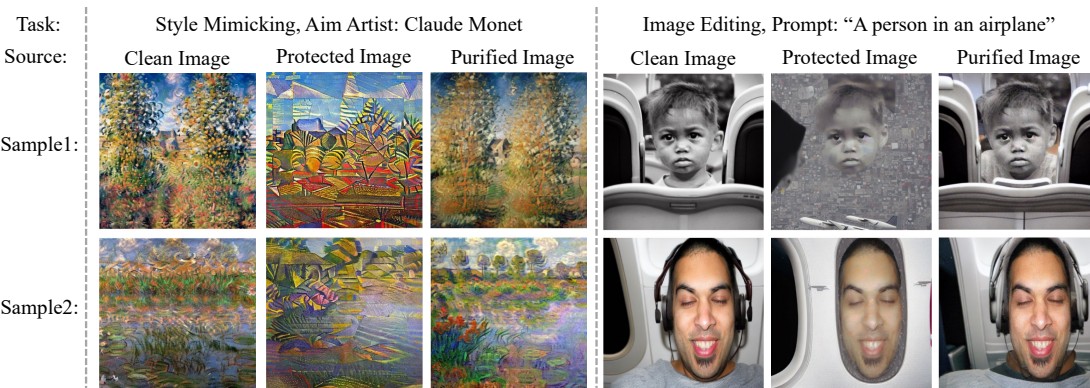

Figure 1: Examples of imperceptible perturbations that are less effective in preventing unauthorized data usage after being purified by IMPRESS. **Left:** Stable diffusion generated images mimicking the style of Claude Monet using clean images, GLAZE (Shan et al., 2023) protected images, and IMPRESS purified images for model fine-tuning. **Right:** images edited by Stable diffusion using the prompt "A person in an airplane" with clean images, PhotoGuard (Salman et al., 2023) protected images, and IMPRESS purified images as model input.

of celebrities downloaded from the Internet to create disinformation (Pranav Dixit, 2023; Li et al., 2022a). Those *unauthorized data usage* without the consent from the original data owner creates severe ethical concerns and causes profound societal harm (Shan et al., 2023).

To prevent unauthorized data usage, some recent studies (Shan et al., 2023; Salman et al., 2023) proposed to add imperceptible perturbations to images such that the latent diffusion model (LDM) (Rombach et al., 2022b), a state-of-the-art diffusion-based image generation model, is not able to leverage perturbed images to generate high-quality samples. For instance, PhotoGuard (Salman et al., 2023) proposed to add imperceptible perturbations to the images such that LDMs cannot generate realistic images when attempting to edit upon the perturbed image. GLAZE (Shan et al., 2023) proposed to add imperceptible perturbations to the artworks of an artist such that the LDMs cannot learn from perturbed images to generate samples that mimic the style of the artist. These works share a common trait: they all aim to add imperceptible perturbations to the original images to protect them from being "correctly" processed by the LDMs without proper authorization.

Although those protection methods (Shan et al., 2023; Salman et al., 2023) show some promising demos on preventing unauthorized data usage, we still lack a systematic understanding of their robustness and practical performances. For instance, since the perturbations added by those methods are imperceptible, we could potentially purify a perturbed image to disable its perturbation.

In this work, we aim to bridge the gap by conducting a systematic examination of those protection methods. We argue that this examination is essential from two aspects. First, the effectiveness of those protection methods under strong, adaptive perturbation purification is unknown. As a result, the examination could help users obtain a better understanding on the effectiveness of those protection methods in different scenarios. Second, the examination could also shed light on the design of new protection methods. The developed system can also be used as an evaluation platform for testing the effectiveness of future protection methods.

Our key observation is that the current "successful" imperceptible perturbations usually lead to a perceptible inconsistency between the original image and the diffusion-reconstructed image (i.e., the image-to-image output of the LDM without given any prompt), which can be used to devise a new optimization strategy for purifying the image and disabling the added perturbation. Specifically, we derive two conditions that the purified image should satisfy: *similarity condition* and *consistency condition*. The similarity condition suggests that the purified image should be visually close to the perturbed image, and thus visually close to the original image (since the perturbation is imperceptible). The consistency condition means that the reconstructed image by a LDM for the purified image should be similar to itself. This condition stems from the observations that the clean image can reconstruct the original image and satisfy such consistency condition while the perturbed image cannot. We defer the detailed derivation of those two conditions to Section 4. Given those two conditions, we respectively design two losses to quantify them, enabling us to formulate the purification of a

perturbed image as an optimization problem. Finally, we solve the optimization problem and obtain the purified image. In summary, our contributions are as follows:

- We have explored and analyzed existing protection methods for unauthorized data usage in diffusion models and pointed out the potential risks. We found that existing methods heavily rely on adding an imperceptible perturbation. Yet such an imperceptible perturbation usually leads to a perceptible inconsistency between the original image and the diffusion-reconstructed image.

- Based on the empirical findings, we propose **IMPRESS**, i.e., **IM**perceptible **P**erturbation **RE**moval **SyS**tem, a new platform for evaluating the effectiveness of imperceptible perturbations as a protective measure by purifying a perturbed image with consistency-based losses.

- We also test the effectiveness of adaptive protection designs which also feature our consistency-based loss upon those existing protection methods. We find that it is difficult to generate effective imperceptible perturbations through such a simple adaptive design, highlighting the need for more advanced methods to prevent unauthorized data usage in LDMs.

## 2 Related Works

**Diffusion-based Image Generation Systems.** Recently, Diffusion Probabilistic Models (DPMs) (Ho et al., 2020) have achieved impressive results in the field of image generation (Croitoru et al., 2023), including applications such as unconditional image generation(Ho et al., 2020; Nichol and Dhariwal, 2021; Song et al., 2020a), text-to-image synthes (Shi et al., 2022; Rombach et al., 2022a; Saharia et al., 2022b), image-to-image translation (Wolleb et al., 2022; Li et al., 2022b; Wang et al., 2022; Zhao et al., 2022; Saharia et al., 2022a), image editing (Avrahami et al., 2022; Meng et al., 2021), and image inpainting(Lugmayr et al., 2022; Bugeau and Bertalmio, 2009). DPMs typically employ a U-Net (Ronneberger et al., 2015) architecture as the underlying neural backbone, which naturally adapts to image-like data (Dhariwal and Nichol, 2021; Ho et al., 2020; Song et al., 2020b; Ronneberger et al., 2015).

**Latent Diffusion Models.** In image generation tasks, training and evaluating DPMs in the original image feature space can lead to low inference speed and high training costs. Recent works have been trying to address this issue by using advanced sampling strategies (Song et al., 2020a; Kong and Ping, 2021; San-Roman et al., 2021), hierarchical approaches (Ho et al., 2022; Vahdat et al., 2021), and feature compression strategy (Rombach et al., 2022b). Among them, latent diffusion model (Rombach et al., 2022b) employs a pre-trained image encoder and decoder, enabling the DPM to work in a compressed, low-dimensional latent space, thus reducing the computational cost of training and accelerating inference speed while maintaining almost the same quality of synthesized images.

**Unauthorized Data Usage in Diffusion Models.** The powerful image generation and editing capabilities of AI have also raised ethical concerns, including malicious image editing (Goodfellow et al., 2014; Mirza and Osindero, 2014; Salimans et al., 2016; Isola et al., 2017; Zhu et al., 2017; Zhang et al., 2017; Karras et al., 2018; Brock et al., 2019; Karras et al., 2019; Rombach et al., 2022b; Ramesh et al., 2022) or training image generation models using unauthorized images from artists and mimicking their styles (Shan et al., 2023). Some works have attempted to address this issue, such as removing certain knowledge from models (Gandikota et al., 2023) or making images unlearnable or uneditable (Huang et al., 2021; Fu et al., 2021; Shan et al., 2023; Salman et al., 2023). However, other studies argue that such imperceptible perturbations are fragile (Radiya-Dixit et al., 2021b).

## 3 Preliminaries

We briefly introduce the preliminaries on Latent Diffusion Models (Rombach et al., 2022b) and existing protection methods (Salman et al., 2023; Shan et al., 2023).

**Latent Diffusion Models (LDMs).** Latent diffusion model (Rombach et al., 2022b) aims to train the diffusion model on a lower-dimensional latent space to generate high-quality samples while saving computation costs. Moreover, it can utilize additional information, such as text, to guide the generation process, enabling it to generate or edit images with guidance from a prompt (e.g., a text). Given an image $\mathbf{x}$, LDM first uses an image encoder $\mathcal{E}$ to embed it into a latent representation, i.e., $\mathbf{z} = \mathcal{E}(\mathbf{x})$ and then perform $T$ forward diffusion steps by progressively adding Gaussian noise $\boldsymbol{\epsilon} \sim \mathcal{N}(\mathbf{0}, \mathbf{I})$, to the latent space. For simplicity, we use $\mathbf{z}_1, \mathbf{z}_2, \cdots, \mathbf{z}_T$ to denote latent representations at each forward diffusion step. Suppose we have a prompt $y$ (e.g., a certain text phrase), a feature extractor $\tau_\theta$

and we denote the embedding of $y$ as $\tau_\theta(y)$. The goal of the LDM is to train a conditional denoising network $\epsilon_\theta$, e.g., a UNet (Ronneberger et al., 2015), to model the conditional distribution $p(\mathbf{z}|y)$ by gradually recovering $\mathbf{z}$ from $\mathbf{z}_T$ based on $\tau_\theta(y)$. Suppose $\epsilon_\theta(\mathbf{z}_t, t, \tau_\theta(y))$ is the Gaussian noise estimated in the $t$-th step. We can train $\epsilon_\theta$ based on the following loss:

$$L_{\text{LDM}} := \mathbb{E}_{\mathcal{E}(\mathbf{x}),y,\epsilon\sim\mathcal{N}(\mathbf{0},\mathbf{I}),t}\left[\|\epsilon - \epsilon_\theta(\mathbf{z}_t, t, \tau_\theta(y))\|_2^2\right]. \tag{3.1}$$

Suppose $\widetilde{\mathbf{z}}$ is the latent representation obtained after $T$ denoising steps. We can use an image decoder $\mathcal{D}$ to obtain the generated image $\widetilde{\mathbf{x}}$ based on $\widetilde{\mathbf{z}}$, i.e., $\widetilde{\mathbf{x}} = \mathcal{D}(\widetilde{\mathbf{z}})$. For simplicity, we denote the end-to-end framework to obtain $\widetilde{\mathbf{x}}$ based on $\mathbf{x}$ and $y$ as $\widetilde{\mathbf{x}} = f_{\text{LDM}}(\mathbf{x}; y)$. Given a prompt, LDMs can be used to generate a sample or edit an image. Note that when there is no prompt $y$, the LDM reconstructs its input image $\mathbf{x}$, i.e., $\mathbf{x}$ would be very similar to $f_{\text{LDM}}(\mathbf{x})$.

**PhotoGuard.** PhotoGuard (Salman et al., 2023) proposes to add carefully crafted perturbations to an image before making it publicly available (e.g., uploading it to the Internet) to raise the cost of malicious image editing. When using LDMs to edit the perturbed images, the output is usually less realistic. Specifically, PhotoGuard proposes two approaches: *encoder attack* and *diffusion attack*.

The goal of *the encoder attack* is to add a perturbation $\boldsymbol{\delta}_{\text{enc}}$ to an image $\mathbf{x}$ such that the image encoder $\mathcal{E}$ produces similar outputs for $\mathbf{x} + \boldsymbol{\delta}_{\text{enc}}$ and a target image $\mathbf{x}_{\text{target}}$ (e.g., a pure gray image or random noise). Formally, $\boldsymbol{\delta}_{\text{enc}}$ can be obtained by solving the following optimization problem:

$$\boldsymbol{\delta}_{\text{enc}} = \underset{||\boldsymbol{\delta}||_\infty \leq \Delta}{\arg\min} \|\mathcal{E}(\mathbf{x} + \boldsymbol{\delta}) - \mathcal{E}(\mathbf{x}_{\text{target}})\|_2^2, \tag{3.2}$$

where $\Delta$ is the $L_\infty$ norm perturbation budget. The noise added to the image could be imperceptible to human eyes when $\Delta$ is small. Thus when LDMs are used to edit $\mathbf{x} + \boldsymbol{\delta}_{\text{enc}}$, the resulting output would be similar to the target image instead. *The diffusion attack*, on the other hand, is more direct and aims to craft a perturbation $\boldsymbol{\delta}_{\text{diff}}$ such that the output of the LDM is similar to $\mathbf{x}_{\text{target}}$:

$$\boldsymbol{\delta}_{\text{diff}} = \underset{||\boldsymbol{\delta}||_\infty \leq \Delta}{\arg\min} \|f_{\text{LDM}}(\mathbf{x} + \boldsymbol{\delta}) - \mathbf{x}_{\text{target}}\|_2^2. \tag{3.3}$$

Compared to the encoder attack which only targets the image encoder, the diffusion attack considers the whole LDM model with prompts, achieving better empirical performance but is less efficient.

**GLAZE.** GLAZE (Shan et al., 2023) aims to add perturbations to the artworks of an artist such that LDMs cannot learn the correct style of the artist from perturbed artworks. Given an image $\mathbf{x}$, GLAZE first chooses a target style $T$ that is sufficiently different from the style of $\mathbf{x}$. Then, GLAZE transfers $\mathbf{x}$ to the target style $T$ using a pre-trained style transfer model $\Omega$. For simplicity, we use $\Omega(\mathbf{x}, T)$ to denote the style-transferred image. Given $\Omega(\mathbf{x}, T)$, GLAZE crafts the perturbation $\boldsymbol{\delta}_{\text{GLAZE}}$ by solving the following optimization problem:

$$\boldsymbol{\delta}_{\text{GLAZE}} = \min_{\boldsymbol{\delta}} \|\mathcal{E}(\Omega(\mathbf{x}, T)), \mathcal{E}(\mathbf{x} + \boldsymbol{\delta})\|_2^2 + \lambda \cdot \max\left(\text{LPIPS}(\mathbf{x}, \mathbf{x} + \boldsymbol{\delta}) - \Delta_L, 0\right), \tag{3.4}$$

where LPIPS (Learned Perceptual Image Patch Similarity) (Zhang et al., 2018) measures user-perceived image distortion, $\Delta_L$ is the perturbation budget, and $\lambda$ is a hyper-parameter adjusting the strength of the LPIPS regularization term. Roughly speaking, GLAZE aims to perturb images (e.g., artworks of an artist) such that a LDM generates samples with the target style instead of the original style when learning from the perturbed images.

# 4 Proposed Method

In this section, we formally study the potential vulnerabilities in existing protection methods and discuss our proposed method to disable perturbations added by those methods.

## 4.1 Analyzing Potential Vulnerabilities of Existing Methods

We notice that existing protection methods (e.g., PhotoGuard and GLAZE) all aim to add imperceptible perturbations such that LDMs cannot "correctly" learn or process the perturbed image for style mimicking or malicious editing. While such perturbations certainly raise the cost of unauthorized data usage, it is not flawless. Specifically, one special use case of LDM is to let it *reconstruct* the image, i.e., let an image $\mathbf{x}$ run through the entire LDM (encoder, forward diffusion, backward diffusion,

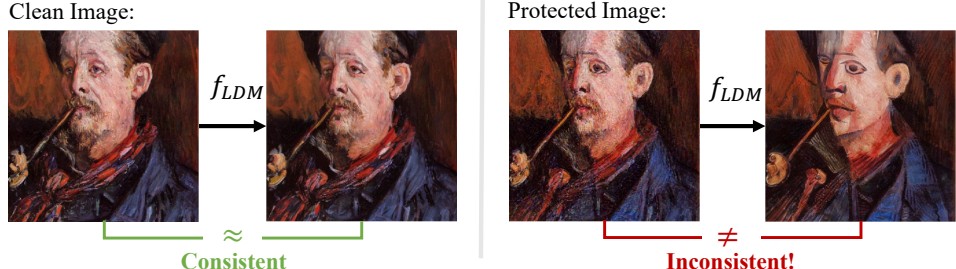

Figure 2: Examples of inconsistencies observed when using LDMs to reconstruct the protected images. **Left**: the diffusion-reconstructed clean image looks similar to the original image; **Right**: images protected by GLAZE (Shan et al., 2023) leads to noticeable inconsistency after reconstruction.

decoder) without any prompt. In such a case, the output of the LDM (denoted by $f_{\text{LDM}}(\mathbf{x})$) should be close to the original input $\mathbf{x}$. However, since the imperceptible perturbations are able to deceive the LDMs (to believe that the image is from another style or that the image's latent embedding is close to a target image), its reconstruction result $f_{\text{LDM}}(\mathbf{x}_{\text{ptb}})$ may no longer be close to the perturbed input $\mathbf{x}_{\text{ptb}}$. Rather, it could become similar to a target image or target style instead.

Figure 2 shows an example painting by Claude Monet and its protected version by GLAZE (Shan et al., 2023). When using LDMs to reconstruct the two images respectively, we can observe that the protected image leads to noticeable inconsistency after reconstruction while the clean image stays roughly the same. In other words, the reconstructed version of the perturbed image is no longer close to itself, rather, it is more close to the targeted style (e.g., Cubism by Picasso). By contrast, a clean image and its reconstructed version do not have such inconsistency. This inspires us to leverage the inconsistency to purify the perturbed image to disable such imperceptible perturbations.

### 4.2 Our Proposed Method

Based on our previous empirical findings, given a perturbed image $\mathbf{x}_{\text{ptb}}$ created by existing protection methods via imperceptible perturbations, we aim to purify the perturbed image to disable its perturbation. We first derive the two conditions that the purified image $\mathbf{x}_{\text{pur}}$ should satisfy, then formulate **IMPRESS**, i.e., **IM**perceptible **P**erturbation **RE**moval **Sy****S**tem, as an optimization problem based on those two conditions and provide our solution to solve it, and finally show our complete algorithm.

**Two conditions for purified image.** The purified image $\mathbf{x}_{\text{pur}}$ should satisfy two conditions: *similarity condition* and *consistency condition*. Next, we respectively derive them.

1. **Similarity condition.** Our ultimate goal is to make sure the purified image $\mathbf{x}_{\text{pur}}$ stays visually close to the original image $\mathbf{x}$, i.e., $\mathbf{x}_{\text{pur}} \approx \mathbf{x}$. Recall that existing methods (Salman et al., 2023; Shan et al., 2023) aim to add an imperceptible perturbation to a given image $\mathbf{x}$ to craft a perturbed image $\mathbf{x}_{\text{ptb}}$ (see Section 3 for details). Therefore, the perturbed image $\mathbf{x}_{\text{ptb}}$ is already close to the original image $\mathbf{x}$. Based on this observation, since we only have access to the perturbed image $\mathbf{x}_{\text{ptb}}$, requiring the purified image $\mathbf{x}_{\text{pur}}$ to stay visually close to the perturbed image $\mathbf{x}_{\text{ptb}}$ would be sufficient to guarantee $\mathbf{x}_{\text{pur}} \approx \mathbf{x}$. This leads to *similarity condition*.

2. **Consistency condition.** As our goal is to find a purified image $\mathbf{x}_{\text{pur}}$ close to the original image $\mathbf{x}$, $\mathbf{x}_{\text{pur}}$ should share some similar traits as $\mathbf{x}$. Our observation in Section 4.1 suggests that a clean image can stay roughly the same after diffusion-reconstruction while the perturbed image cannot. Therefore, to make $\mathbf{x}_{\text{pur}}$ close to $\mathbf{x}$, we also require the purified image $\mathbf{x}_{\text{pur}}$ to be able to reconstruct itself using LDMs, i.e., $\mathbf{x}_{\text{pur}} \approx f_{\text{LDM}}(\mathbf{x}_{\text{pur}})$. This leads to *consistency condition*.

**Formulating IMPRESS as an optimization problem.** Our key idea is to define two losses that respectively quantify the similarity condition and consistency condition.

1. **Quantifying similarity condition.** Our similarity condition means that $\mathbf{x}_{\text{pur}}$ would be visually close to $\mathbf{x}_{\text{ptb}}$. Following previous studies (Shan et al., 2023), we use LPIPS (Learned Perceptual Image Patch Similarity) (Zhang et al., 2018) to quantify the visual difference between $\mathbf{x}_{\text{pur}}$ and $\mathbf{x}_{\text{ptb}}$. Formally, we have the loss $\max(\text{LPIPS}(\mathbf{x}_{\text{pur}}, \mathbf{x}_{\text{ptb}}) - \Delta_L, 0)$, where $\Delta_L$ is the perceptual perturbation budget. We note that LPIPS is widely adopted in various fields (Cherepanova et al.,

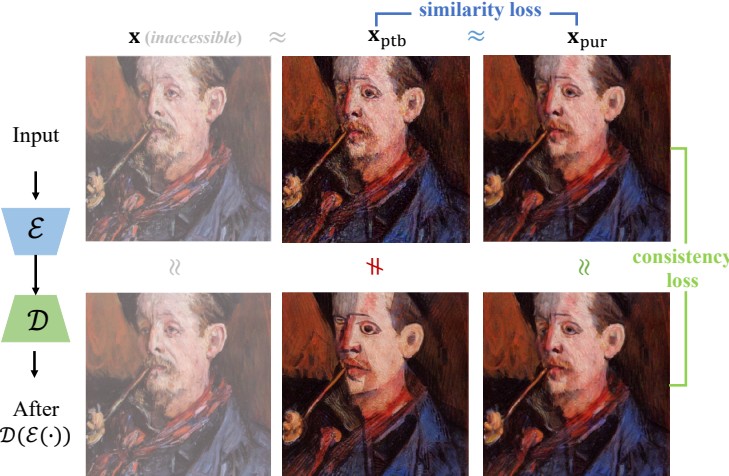

Figure 3: A sketch of our method consists of two losses: the *similarity loss* and the *consistency loss*. The *similarity loss* requires the purified image $\mathbf{x}_{\text{pur}}$ to stay close to $\mathbf{x}_{\text{ptb}}$ while the *consistency loss* requires the encoder/decoder reconstructed image to stay close $\mathbf{x}_{\text{pur}}$ itself.

    2021; Laidlaw et al., 2020; Rony et al., 2021) as it provides a measure of human-perceived image distortion. Compared to $\ell_p$-norm constraint, LPIPS offers more flexible constraint ranges. If the modified image still appears similar to the original image from a human perspective, it allows for larger changes in certain parts of the image, which aligns with our goal of preserving as much semantic information from $\mathbf{x}_{\text{ptb}}$ as possible.

2. **Quantifying consistency condition.** Our consistency condition implies that $\mathbf{x}_{\text{pur}} \approx f_{\text{LDM}}(\mathbf{x}_{\text{pur}})$. Intuitively, we can define the consistency loss as $||\mathbf{x}_{\text{pur}} - f_{\text{LDM}}(\mathbf{x}_{\text{pur}})||_2^2$. However, it is very challenging to directly optimize this loss as it involves the complicated diffusion process (i.e., gradually adding and removing Gaussian noise). To address the challenge, we simplify the loss by removing the diffusion process in it. In particular, we define the loss $||\mathbf{x}_{\text{pur}} - \mathcal{D}(\mathcal{E}(\mathbf{x}_{\text{pur}}))||_2^2$ to quantify the consistency condition, where $\mathcal{E}$ and $\mathcal{D}$ are the image encoder and decoder in the LDM. Note that $||\mathbf{x}_{\text{pur}} - \mathcal{D}(\mathcal{E}(\mathbf{x}_{\text{pur}}))||_2^2$ can approximate $||\mathbf{x}_{\text{pur}} - f_{\text{LDM}}(\mathbf{x}_{\text{pur}})||_2^2$ as the goal of the diffusion process (without any prompt) is to reconstruct latent representation. Moreover, such approximation enables us to solve the optimization problem efficiently.

Combining the above two losses together, our final optimization problem is as follows:

$$\min_{\mathbf{x}_{\text{pur}}} \underbrace{||\mathbf{x}_{\text{pur}} - \mathcal{D}(\mathcal{E}(\mathbf{x}_{\text{pur}}))||_2^2}_{\text{consistency loss}} + \alpha \cdot \underbrace{\max(\text{LPIPS}(\mathbf{x}_{\text{pur}}, \mathbf{x}_{\text{ptb}}) - \Delta_L, 0)}_{\text{similarity loss}}, \tag{4.1}$$

where $\alpha$ is a hyper-parameter to balance the two losses. Given a perturbed image $\mathbf{x}_{\text{ptb}}$, the solution to the optimization problem in Eq. (4.1) is the purified image $\mathbf{x}_{\text{pur}}$. We use projected gradient descent to solve the optimization problem efficiently. In practice, we initialize $\mathbf{x}_{\text{pur}}$ by $\mathbf{x}_{\text{ptb}}$ plus an extra small Gaussian perturbation similar as in Madry et al. (2018). A summary of our proposed method is shown in Algorithm 1 in the Appendix.

## 5 Experiments

In this section, we conduct comprehensive experiments to demonstrate the effectiveness of the proposed method and verify its capability to disable imperceptible perturbations introduced by existing protection methods. Specifically, we evaluate against two protection methods on the specific tasks for which these methods were proposed: *style mimicking* and *malicious editing of images*.

### 5.1 Experimental Settings

**Style Mimicking**     We conduct all the *style mimicking* experiments on the WikiArt dataset (Saleh and Elgammal, 2015). For GLAZE[1] (Shan et al., 2023), we essentially followed the settings of the

---

[1]Experiments reproduced based on the 3rd version of the GLAZE paper on Arxiv (11 Apr 2023).

original paper and reproduced the code: given a victim artist $V$, we randomly select 124 of their works. We use ViT+GPT2 model to generate a title for each artwork and attach the victim artist $V$'s name to the generated title as our prompt $y$. Next, we randomly draw 24 works from these 124 art pieces as the style imitation training set, and the remaining 100 as the test set. We fine-tune a pre-trained LDM on the training set, and then generate new artworks mimicking the style of the victim artist by the prompt $y$.

**Malicious Editing of Images**  For *malicious editing of images*, we utilize the code and settings made public by Photoguard (Salman et al., 2023). The purpose of *malicious editing* is to modify the background of a photo following the given malicious prompt. We conduct our experiments on the Helen (Le et al., 2012) dataset for this task. We selected the 80 images with the smallest face-to-image ratio from the Helen dataset and their corresponding face masks. Subsequently, we use Photoguard to generate protective noise for these images. Then, we attempt to modify the images using a latent diffusion model. We generate noise using the most powerful *Diffusion attack-L2* method in Photoguard.

**Baselines**  To show the effectiveness of IMPRESS is obtained through our consistency-based loss design, we also consider three other simple post-processing baselines: JPEG compression of the image, adding Gaussian noise to the image, and resizing the image. For JPEG compression, we set the quality factor as 15%. For adding Gaussian noise, we set the mean as 0 and the variance as 0.15. For the image resizing method, we first resize the $512 \times 512$ image to $256 \times 256$, then resize it back to $512 \times 512$.

## 5.2 Experimental Results on Style Mimicking

Table 1: Comparison of style mimicking performances when using fine-tuning data from clean images, GLAZE-protected images and post-processed protected images by our method and baselines. A higher accuracy suggests that the generated image style is close to the artist to be mimicked.

| Metric | Clean | Protected | Post-processed | | | |
|---|---|---|---|---|---|---|
| | | | JPG | Noise | Resize | Ours |
| CLIP classifier Acc | $90.8 \pm 3.2$ | $42.5 \pm 8.3$ | $64.9 \pm 7.6$ | $47.9 \pm 7.0$ | $66.6 \pm 3.4$ | **$87.0 \pm 6.7$** |
| Diffusion classifier Acc | $95.4 \pm 3.1$ | $42.9 \pm 9.5$ | $67.6 \pm 9.0$ | $40.9 \pm 9.8$ | $68.0 \pm 9.5$ | **$82.3 \pm 8.3$** |

In Table 1, we present the experimental results for the task of *Style Mimicking*. To conduct the experiment, we fine-tune the pre-trained LDMs using clean data, GLAZE (Shan et al., 2023) protected data, and the post-processed data by our method as well as other baselines. We then use those fine-tuned models to generate new images mimicking the style of a certain artist and evaluate whether the generated image can successfully mimic the style of the artist. We consider two metrics for evaluating the generated image styles by constructing two image style classifier:

1. **the CLIP classifier accuracy.** Following the GLAZE paper (Shan et al., 2023), we use the pre-trained CLIP model (Radford et al., 2021) to build a style classier to measure whether the generated images belong to the genre of victim artist $V$. We treat the genre classification provided in the WikiArt dataset as the ground truth label, and use the intersection of the 27 historical genres and 13 digital genres from WikiArt, as candidate labels for the CLIP model.

2. **the Diffusion classifier accuracy.** We consider another classifier that builds directly upon diffusion models. The Diffusion Classifier (Li et al., 2023) is a Zero-Shot image classification method, which uses conditional density estimates of images in pre-trained Diffusion models to construct Zero-Shot image classifiers. In our experiments, we used the same set of candidate labels (39 art genres) to build a diffusion classifier for style classification.

For each generated image, we use both the CLIP classifier and Diffusion classifier to evaluate whether the style mimicking succeeds (whether the victim artist $V$'s style falls into the top-3 classification results). Typically, all protection methods aim to lower the classifier accuracy as much as possible to prevent style mimicking from happening if the data usage is unauthorized. As can be seen from Table 1, the images generated using clean data samples achieve an average accuracy of 90.8% on the CLIP classifier and 95.4% on the Diffusion classifier. While the generated images using the protected data

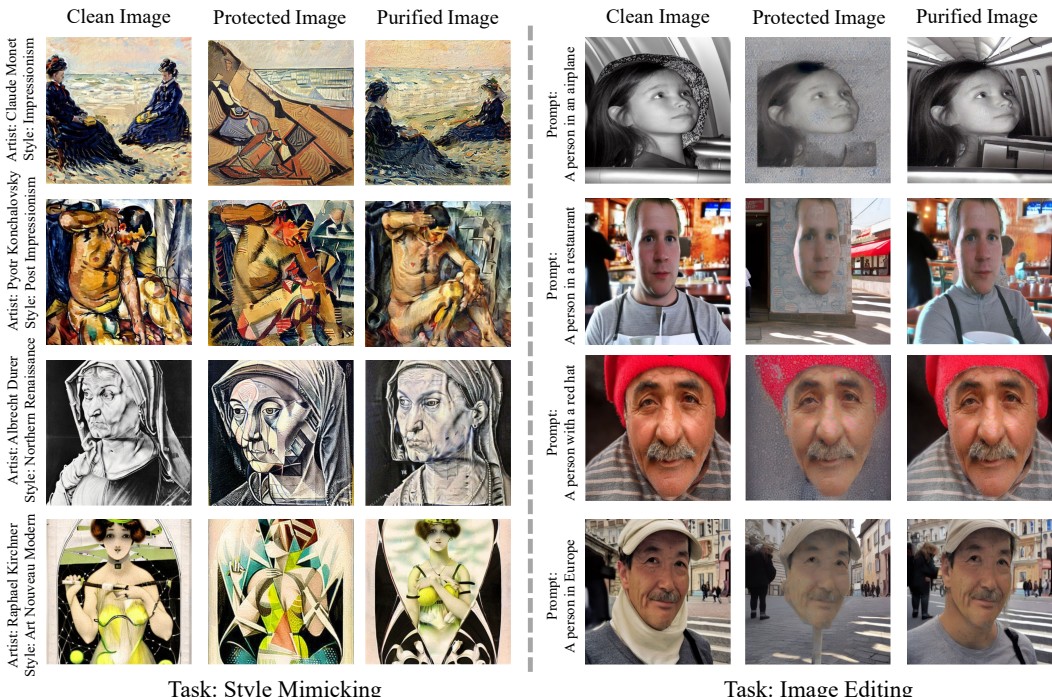

Figure 4: The experimental results of IMPRESS. **Left:** 4 groups of samples in Style Mimicking, each group of samples is generated by Stable diffusion when using clean images, GLAZE (Shan et al., 2023) protected images, and our IMPRESS purified images for model fine-tuning. **Right:** 4 groups of samples in Malicious Editing of Images, each group of samples is obtained by editing the clean image, PhotoGuard (Salman et al., 2023) protected image, and our IMPRESS purified image.

samples certainly lower the accuracy to $\sim 42\%$, after our purified process, the generated images can again achieve an average accuracy of 87.0% and 82.3% on the CLIP classifier and Diffusion classifier respectively, close to the case of clean data samples. By contrast, all the other post-processing baselines only achieved limited improved accuracy, clearly outperformed by our IMPRESS method. We also present some sample results in Figure 4. From Figure 4 (left), we can clearly observe that although the GLAZE method can successfully mislead the LDM when directly used, after using IMPRESS to purify the protected image, we can still exploit the remaining information.

## 5.3 Experimental Results on Malicious Editing

Table 2: Comparison of malicious editing performances with input data from clean images, Photoguard-protected images and post-processed protected images by our method and baselines. The higher score indicates better alignment with the clean images.

| Metric | Protected | Post-processed | | | |
| --- | --- | --- | --- | --- | --- |
| | | JPG | Noise | Resize | Ours |
| SSIM | $0.41 \pm 0.06$ | $\mathbf{0.52 \pm 0.07}$ | $0.18 \pm 0.05$ | $0.49 \pm 0.08$ | $0.49 \pm 0.08$ |
| PSNR | $14.24 \pm 1.57$ | $15.51 \pm 1.69$ | $11.06 \pm 0.94$ | $14.94 \pm 1.59$ | $\mathbf{15.61 \pm 2.03}$ |
| VIF-p | $0.11 \pm 0.03$ | $0.15 \pm 0.04$ | $0.06 \pm 0.02$ | $0.14 \pm 0.05$ | $\mathbf{0.16 \pm 0.04}$ |

For the *Malicious Editing of Images* task, we use a pre-trained LDM to edit images following the guidance of certain prompts. Again we consider input images from clean images, images protected by Photoguard (Salman et al., 2023), and the post-processed images by our method as well as other baselines. We followed the same settings as in the Photoguard paper (Salman et al., 2023), using SSIM (Wang et al., 2004), PSNR, and VIF-p (Sheikh and Bovik, 2006) as image similarity algorithms to measure the fidelity of the generated images. We calculate similarity scores between the edited

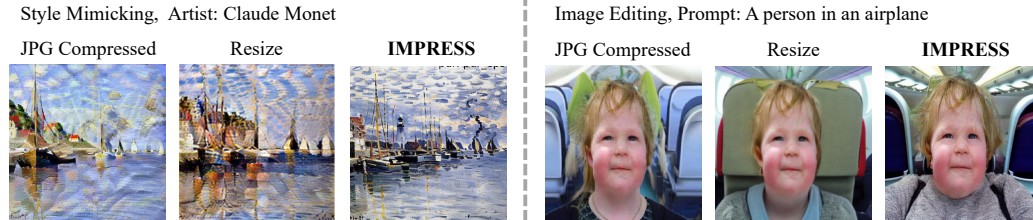

Figure 5: Comparison of different post-processing methods on style mimicking and image editing tasks.

clean images and the edited protected images, as well as the post-processed images. Higher scores indicate results that are closer to the edited clean images.

According to Table 2, our proposed IMPRESS method achieved the best performance on the PSNR and VIF-p metrics, scoring 15.61 and 0.16, respectively, suggesting the edited images are still close to true images. We also obtained a score of 0.49 on the SSIM metric, which is the second-highest score. Note that in malicious editing, even simple image transformation methods can achieve relatively good results on those comparison-based metrics, which has also been pointed out in Salman et al. (2023) and Sandoval-Segura et al. (2023). This phenomenon confirms our point of view: existing image protection technologies are fragile. In Figure 4 (right), we show some real examples that while directing applying PhotoGuard can certainly cause the edited images less realistic, after being purified by our IMPRESS, the malicious editing would still go smoothly with no obvious artifact affecting the image fidelity.

In Figure 5, we also show some comparison with other post-processing baselines on both style mimicking and image editing. We can observe that IMPRESS largely outperforms other baselines for style mimicking while on image editing, all baselines can achieve decent performances in removing the perturbations. This is also consistent with our previous quantitative results, suggesting the imperceptible perturbations on the image editing task is significantly more fragile.

## 5.4 Adaptive Image Protection with Consistency-based Losses

In this section, we explore whether it is possible to leverage our consistency-based losses for building even better protection methods that could survive our IMPRESS method and successfully prevent unauthorized data usage in diffusion models. Specifically, since we know that keeping $\mathbf{x}$ as consistent as possible with $\mathcal{D}(\mathcal{E}(\mathbf{x}))$ is important, we add this regularization term to the optimization function of the existing protection method design. Take GLAZE as an example and we have the following optimization problem:

$$\min_{\boldsymbol{\delta}} \|\mathcal{E}(\Omega(\mathbf{x}, T)), \mathcal{E}(\mathbf{x} + \boldsymbol{\delta})\|_2^2 + \lambda \cdot \max\left(\text{LPIPS}(\mathbf{x}, \mathbf{x} + \boldsymbol{\delta}) - \Delta_L, 0\right) + \beta \cdot \|(\mathbf{x} + \boldsymbol{\delta}) - \mathcal{D}(\mathcal{E}(\mathbf{x} + \boldsymbol{\delta}))\|_2^2,$$

The coefficient of the added consistency regularization term is denoted as $\beta$. A meticulous tuning of the weight of the adaptive loss term has been carried out, as demonstrated in Table 3. We have modulated the weight of the adaptive loss term extensively, allowing it to comprise approximately 10% to 95% of the total loss. Our observations reveal that a minuscule proportion of the adaptive loss term in the entire loss leads the experimental results to mirror those of conventional image protection techniques, whereas an excessive proportion adversely impacts the performance of image protection.

The most pronounced effect of the adaptive protection technology is observable when $\beta = 40$, and we show the experimental results in Table 4 and 5 based on that. However, even under this condition, IMPRESS manages to attain around 80% accuracy by successfully mitigating the adaptive protective noise. Our experiments indicate that the adaptive design does not genuinely offer robust protection, since IMPRESS can still adeptly eliminate the protective noise, allowing the successful execution of style mimicking tasks, it is conjectured that attaining an optimal equilibrium among the three objective terms may pose a significant challenge.

To enhance the balance of competing objectives, we have incorporated the Orthogonal Projected Gradient Descent (O-PGD) approach by Gowal et al. (2020) into our methodology, as delineated

Table 3: CLIP classifier Acc for different weight of the adaptive loss term on style mimicking tasks

| Training data | $\beta = 1$ | $\beta = 10$ | $\beta = 40$ | $\beta = 100$ | $\beta = 1000$ |
|---|---|---|---|---|---|
| Protected | 42.5 | 41.0 | 40.8 | 47.1 | 77.1 |
| IMPRESS | 85.6 | 84.6 | 81.9 | 85.7 | 82.1 |

Table 4: Experimental Results of Adaptive Protection on Style Mimicking Tasks

| Metric | Clean | Protect Without Adapt | | With Adapt | |
|---|---|---|---|---|---|
| | | Protected | Purified | Protected | Purified |
| CLIP classifier Acc | $90.8 \pm 3.2$ | $42.5 \pm 8.3$ | $87.0 \pm 6.7$ | $40.8 \pm 9.1$ | $81.9 \pm 5.5$ |
| Diffusion classifier Acc | $95.4 \pm 3.1$ | $42.9 \pm 9.5$ | $82.3 \pm 8.3$ | $44.1 \pm 9.3$ | $84.0 \pm 6.6$ |

Table 5: Experimental Results of Adaptive Protection on Malicious Editing Tasks

| Metric | Protect Without Adapt | | With Adapt | |
|---|---|---|---|---|
| | Protected | Purified | Protected | Purified |
| SSIM | $0.41 \pm 0.06$ | $0.49 \pm 0.08$ | $0.42 \pm 0.04$ | $0.52 \pm 0.05$ |
| PSNR | $14.24 \pm 1.57$ | $15.61 \pm 2.03$ | $14.54 \pm 1.74$ | $15.90 \pm 1.23$ |
| VIF-p | $0.11 \pm 0.03$ | $0.16 \pm 0.04$ | $0.14 \pm 0.04$ | $0.16 \pm 0.03$ |

Table 6: Experimental results of adaptive defense after using O-pgd. In parentheses are the results of the adaptive defense experiments reported in the paper (without the use of O-pgd)

| Metric | Clean | Protect Without Adapt | | With O-pgd Adapt | |
|---|---|---|---|---|---|
| | | Protected | Purified | Protected | Purified |
| CLIP classifier Acc | 90.8 | 42.5 | 87.0 | 48.0 (40.8) | 83.9 (81.9) |
| Diffusion classifier Acc | 95.4 | 42.9 | 82.3 | 47.8 (44.1) | 84.(84.0) |

in Table 6 in the pdf. However, it's discernible that the utilization of O-PGD doesn't substantially bolster the efficacy of the adaptive method.

Based on our experimental observations, we hypothesize that the subpar performance of the adaptive method stems primarily from two intertwined complexities:1) The introduced supplementary loss term complicates the optimization of the loss function. For GLAZE method, the loss function of the adaptive method has three components. Optimizing three different loss components is typically challenging, and striking a balance among these components remains an intricate task. 2) The loss term employed for the adaptive method might clash with the inherent loss term of image protection techniques. The purpose of image protection technologies is to induce semantical differences between generated images and original ones. Our image purification approach aims to make the image maintains its semantic properties (consistency after reconstruction), potentially leading to underlying optimization conflicts. These results suggest that designing adaptive image protection techniques specifically for IMPRESS might inherently be challenging.

# 6   Conclusion

In this work, proposed IMPRESS, a unified platform to evaluate the effectiveness of imperceptible perturbations as a protective measure. We demonstrated that it is difficult for the current imperceptible perturbations-based protection methods to prevent certain images from being correctly learned or processed by diffusion models as those imperceptible perturbations could be potentially removed with consistency-based losses. The proposed IMPRESS platform offers a comprehensive evaluation on several contemporary protection methods and can be used as an evaluation platform for future protection methods.

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

# A Algorithm of IMPRESS

The following outlines the algorithmic process of our proposed IMPRESS method for the removal of protective noise.

---
**Algorithm 1** IMPRESS

---

**Input:** Image encoder $\mathcal{E}$, image decoder $\mathcal{D}$; hyperparameters $\alpha$,$\eta$, $\Delta_L$.

1: Initialize $\mathbf{x}_{\text{pur}} = \mathbf{x}_{\text{ptb}} + \mathcal{N}(\mu, \sigma^2\mathbf{I})$
2: **for** $iter = 1, 2, \cdots, N$ **do**
3:     $\mathcal{L}_{\text{sim}} = ||\mathbf{x}_{\text{pur}} - \mathcal{D}(\mathcal{E}(\mathbf{x}_{\text{pur}}))||_2^2$
4:     $\mathcal{L}_{\text{lpips}} = \max(\text{LPIPS}(\mathbf{x}_{\text{pur}}, \mathbf{x}_{\text{ptb}}) - \Delta_L, 0)$
5:     $\mathcal{L} = \mathcal{L}_{\text{sim}} + \alpha\mathcal{L}_{\text{lpips}}$
6:     $\mathbf{x}_{\text{pur}} \leftarrow \mathbf{x}_{\text{pur}} - \eta\nabla_{\mathbf{x}}\mathcal{L}$
7:     $\mathbf{x}_{\text{pur}} = \text{Clip}(\mathbf{x}_{\text{pur}}, \min = -1, \max = 1)$
8: **end for**
9: Return $\mathbf{x}_{\text{pur}}$

---

# B Detail of Experiment

In this section, we have provided a detailed introduction to our experimental setup. The random seed for all experiments is set to 0, and our experiment code is publicly available in `https://github.com/AAAAAAsuka/Impress`.

**Style Mimicking** Since we do not have access to unreleased artworks, we conduct all the *style mimicking* experiments on the WikiArt dataset (Saleh and Elgammal, 2015). The WikiArt dataset contains paintings from 195 famous artists throughout history. The dataset consists of 42,129 training images and 10,628 testing images. WikiArt categorizes all artists into 27 different genres (e.g., Impressionism, Cubism). we remove artists from the WikiArt dataset that the CLIP model cannot accurately classify (classification accuracy less than 80%). Then, we select nine artists with the most artworks among the remaining artists, eight of which are used for testing the model's performance, and the remaining artist serves as the validation set for adjusting hyperparameters.

For GLAZE (Shan et al., 2023), we basically follow the original paper's setting: given a victim artist $V$, we randomly select 124 of their works. Then, we use the ViT+GPT2 model to generate a title for each work and attach the victim artist $V$'s name to the generated title as the prompt $y$. Next, we randomly draw 24 works from these 124 art pieces as the style imitation training set, and the remaining 100 as the test set. Afterward, we use the *stable-diffusion-2-1-base* model released on Hugging Face as the initial pre-trained model weights and fine-tune it on the training set. After obtaining the fine-tuned LDM, we generate imitation works in the style of the victim artist using the prompts from the training set. Finally, we selected 8 artists and randomly extracted 24 pieces for fine-tuning the diffusion model and 100 pieces for validation, totaling 800 evaluation samples.

For fine-tuning the diffusion model, we use the fine-tuning script released with the model, training for 500 steps on the training set with a learning rate of $5 \times 10^{-6}$. For GLAZE, we choose a perturbation budget of $p = 0.05$, regularization coefficient $\alpha = 30$, and train for 500 steps using the Adam optimizer with a learning rate of $10^{-2}$. For our method, we set the perturbation budget to $p = 0.1$, regularization coefficient $\alpha = 0.1$, and train for 3,000 steps using the Adam optimizer with a learning rate of $10^{-2}$.

**Malicious Editing of Images** We use the Helen (Le et al., 2012) dataset for the experiments in the *Malicious Editing of Images* task. Helen is an annotated facial image dataset, consisting of 2,330 images sourced from Flickr. The annotations include facial landmarks and masks of facial features such as eyes, nose, etc. The dataset also includes masks for the entire face.

For *malicious editing of images*, We utilize the code and settings made public by Photoguard(Salman et al., 2023) for image protection. The purpose of the *Malicious Editing of Images* task is to modify the background of a photo. Hence, we selected the 80 images with the smallest face-to-image ratio from the Helen dataset and their corresponding face masks (the sample size of Photoguard experiment

is 60). Subsequently, we use Photoguard to generate protective noise for these images. Then, we attempt to modify the images using a diffusion model with the prompt "A person in an airplane".

We generate noise using the most powerful *Diffusion attack-L2* method in Photoguard, and all hyperparameters adhere to the settings in the open-source code. We use the *runwayml/stable-diffusion-inpainting* model released on Hugging Face to modify images, with a guidance strength of 7.5 and 100 diffusion steps.For our method, we set the perturbation budget to $p = 0.1$, regularization coefficient $\alpha = 10^{-2}$, and train for 1,000 steps using the Adam optimizer with a learning rate of $5 \times 10^{-3}$.

## C    More Experimental Results

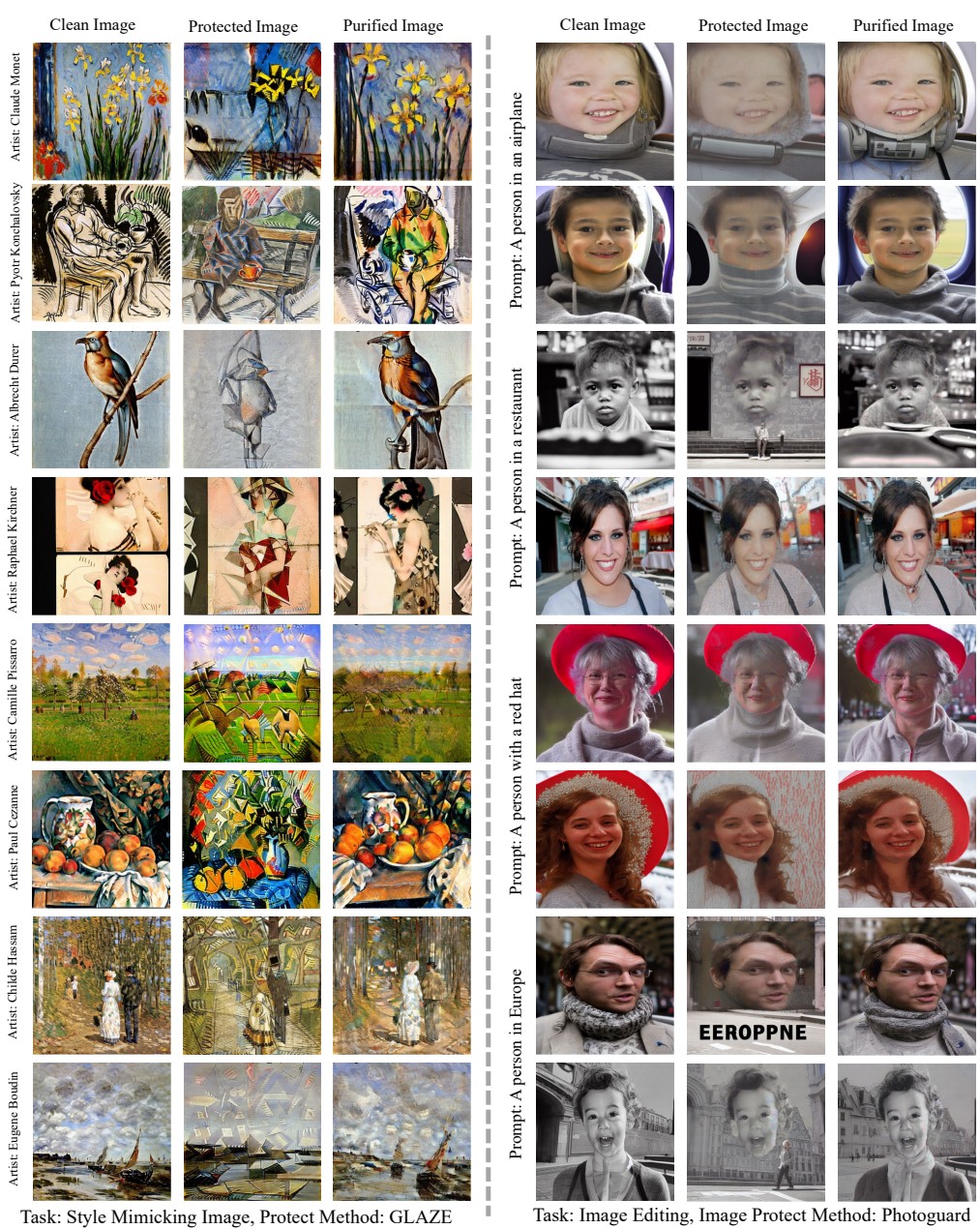

Figure 6: More Experimental Results

# D    Ablation Study

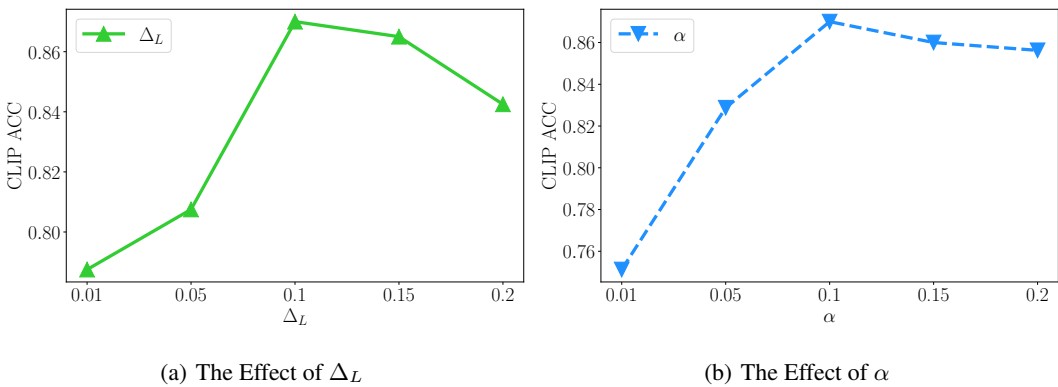

(a) The Effect of $\Delta_L$                    (b) The Effect of $\alpha$

Figure 7: Ablation Study of *Style Mimicking*

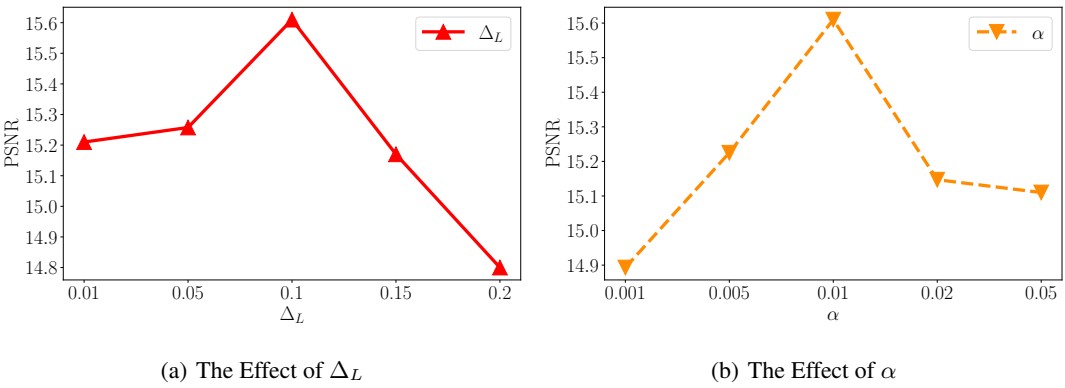

(a) The Effect of $\Delta_L$                    (b) The Effect of $\alpha$

Figure 8: Ablation Study of *Malicious Editing of Images*

In this section, we analyze the effects of two hyperparameters in our method: $\alpha$ and $\Delta_L$. Among them, $\alpha$ is the balance weight in the entire optimization objective of Impress, and $\Delta_L$ is the threshold of the LPIPS regularization term. Here, we use the CLIP evaluation indicator in the *Style Mimicking* task and the PSNR indicator in the *Malicious Editing of Images* task to evaluate the impact of hyperparameters. The evaluation results are shown in Figure 7 and Figure 8. For the threshold $\Delta_L$ of the LPIPS regularization term, by observing Figure 7(a) and Figure 8(a), we can see that when $\Delta_L$ is too large, the LPIPS term loses its constraining effect. This may cause the purification process to make excessive modifications to the original image, thereby destroying the semantic information such as style/content in the original image, resulting in poor performance. When $\Delta_L$ is too small, the LPIPS regularization term dominates the optimization process. This causes the purified image to be too similar to the protected image, retaining some protective noise, thereby affecting performance. Similarly, by observing Figure 7(b) and Figure 8(b), we can find: 1) When the balance weight $\alpha$ is too large, it will cause the weight of the LPIPS term to be too large, causing similar results to when $\Delta_L$ is too small, that is, retaining too much protective noise, thus leading to a decrease in purification effect. When $\alpha$ is too small, the LPIPS term loses its constraining effect.

At the same time, by comparing Figure 7 and Figure 8, we found that the Photoguard method is more sensitive to hyperparameters. We believe this may be because the disturbance is constrained by $L_2$ in Photoguard, which leads to some obvious point-like artifacts on the image. These point-like artifacts may play a "trigger" role. When they cannot be effectively removed, the purification effect will decrease.

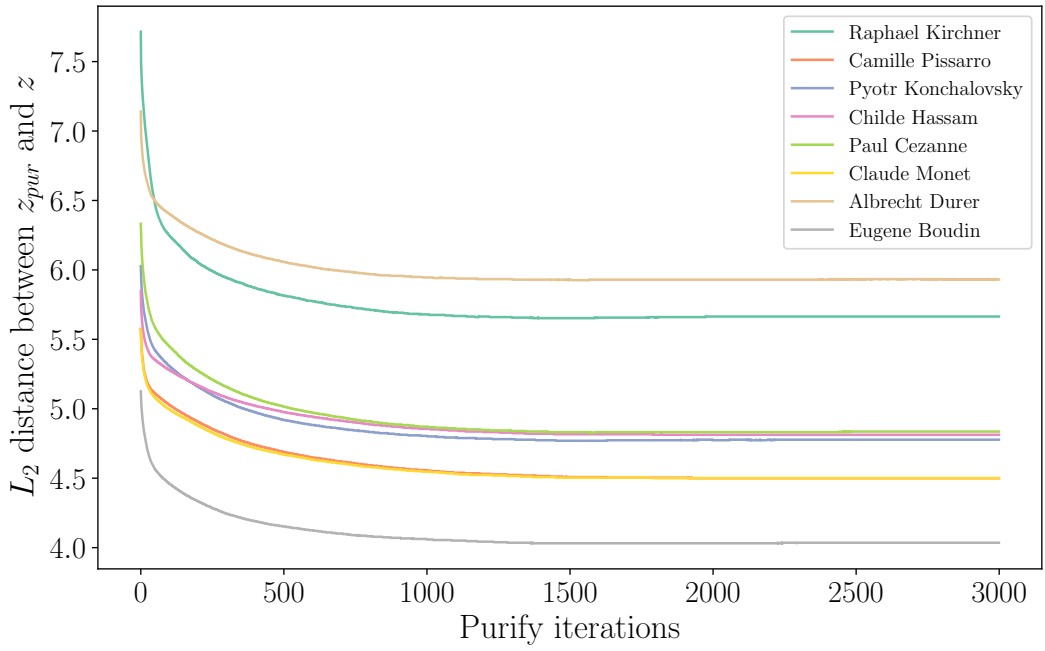

Figure 9: $L_2$ distance between $\mathbf{z}_{\mathrm{pur}}$ and $\mathbf{z}$

## E  A Metric-based validity study

To validate the effectiveness of our method, we calculated the distance of the embedding $\mathbf{z}_{\mathrm{pur}}$ of the purified image in the diffusion model relative to the embedding $\mathbf{z}$ of the clean image in the *Style Mimicking* task during the purification process. Please note that in the entire purification process of IMPRESS, we cannot get the clean image, that is, we do not know $\mathbf{z}$.

We plotted the results in Figure 9. For a protected image, we recorded the embedding $\mathbf{z}_{\mathrm{pur}}$ after each step in the purification process using IMPRESS, totaling 3000 steps (consistent with our experimental setup). Then we normalize $\mathbf{z}$ and all $\mathbf{z}_{\mathrm{pur}}$, and then calculate the $L_2$ distance between $\mathbf{z}_{\mathrm{pur}}$ after each step and $\mathbf{z}$, thereby obtaining the $L_2$ distance between $\mathbf{z}_{\mathrm{pur}}$ after each step and $\mathbf{z}$ during the purification process of this image. Finally, we average the results obtained from all pictures selected from each artist. We plotted the average value of each artist in Figure 9 as a separate curve.

As can be seen, for all artists, the $L_2$ distance between $\mathbf{z}_{\mathrm{pur}}$ after each step in the purification process and $\mathbf{z}$ gradually decreases. This means that the feature vector of the purified image in the model is more similar to the clean image, which also means that our method effectively removes the protective noise and is effective for all kinds of different types of artworks.

## F  Real-World Style Mimicking Service

We tested our method on a real-world image generation service website. *scenario.com* is a image generation service provider that allows users to upload a set of images and mimic the style of these images, and then generate images with similar styles based on the prompts provided by the user. We randomly selected the works of 4 artists in our experiment as the targets for imitation, then used *scenario.com* to mimic clean images, images protected by GLAZE, and images purified by IMPRESS respectively, and generated two sets of new images for each. The training data used in the imitation process is consistent with the data used in the default parameter settings in our experiment, and the test results are shown in Figure 10. As can be seen, GLAZE effectively protects the images on real-world services. The style of the image protected by GLAZE is completely different from the original image, and the content of the picture is severely damaged, sometimes it is difficult to identify the content of the image. On the other hand, new images generated by imitating IMPRESS purified images are highly consistent in style and content with those generated by imitating clean images, which implies that our method can effectively remove protective noise.

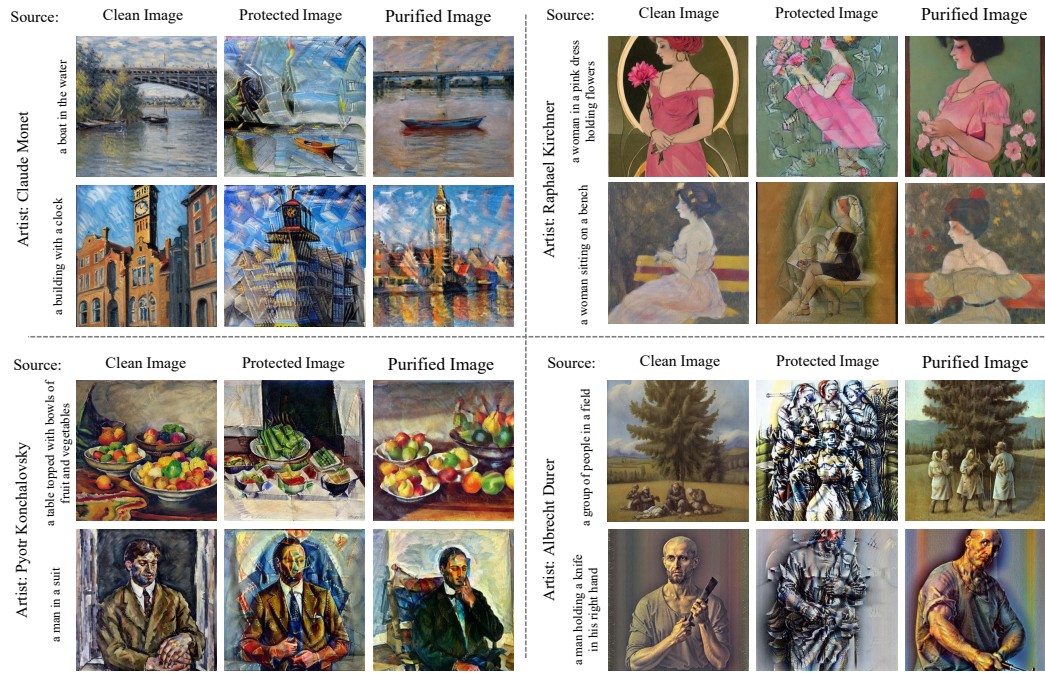

Figure 10: The images generated using the image style imitation service provided by *scenario.com* to mimic clean images, images protected by GLAZE, and images purified by IMPRESS. The prompt used to generate each group of images is shown on the left side of each group. All images are generated using the default parameters provided by *scenario.com*.

## G Impact of IMPRESS on Clean Images

After testing, our method does not affect clean data. We conducted an experiment where IMPRESS was directly applied to clean images. For the *Style Mimicking* task, the ACC of the CLIP Classifier was 91.2%, and the ACC of the Diffusion Classifier was 92.5%, both of which were close to the results obtained from fine-tuning the model directly using clean data. For the *Malicious Editing of Images* task, SSIM, PSNR, and VIF-p were respectively 0.59, 16.70, and 0.23, significantly higher than other experiments. Figure 11 shows the images obtained by using IMPRESS to purify clean images and the results of using clean images directly in the two tasks, showing that our method hardly impacts clean images.

## H Explanation of Differences in GLAZE Experimental Results

In the original GLAZE paper, the selection method of the target style $T$ is to randomly choose a style that is significantly different from the victim artist $V$, but the paper does not provide specific style selection data and hyperparameter settings. We attempted to reproduce the target style selection method proposed in the original GLAZE paper as much as possible and conducted multiple random experiments. Using the same data and default hyperparameters, the average accuracy of the image generated by the Protected image as training data in the CLIP classifier is 59.8% when using the target style selection method of GLAZE, while the average result after purification by IMPRESS is 87.9%. We found that the GLAZE method is very sensitive to the target style $T$. For a particular artist, a reasonable target style can make the average accuracy of the GLAZE method in the CLIP classifier less than 10%.

To ensure the consistency and reproducibility of the experiment, we choose to use "Cubism by Picasso" as the target style in our tests, because "Cubism by Picasso" is one of the target style prompts used by GLAZE in the example results shown in the paper, and a clear protective effect can be seen. With the target style unified, "Cubism by Picasso" is also the best performing target style. The specific results for each artist are shown in Table 7.

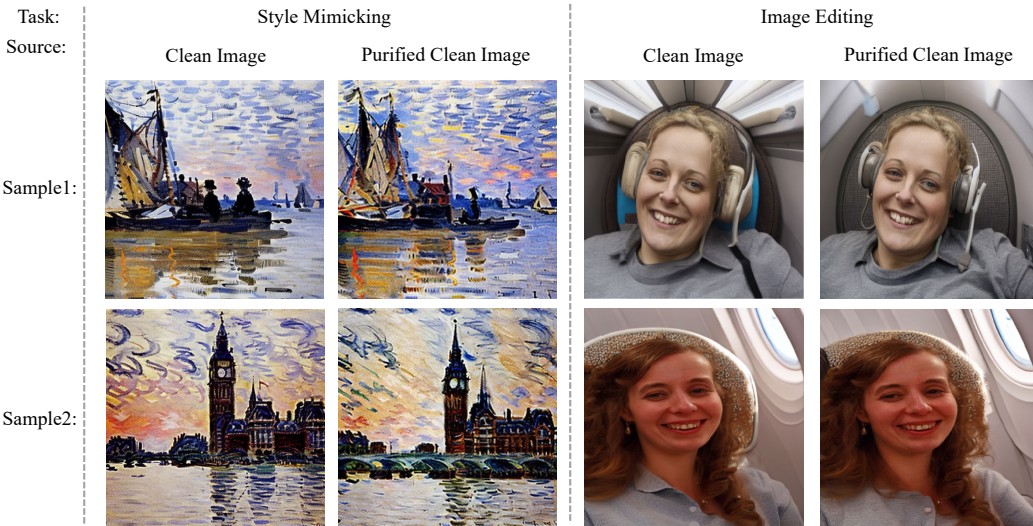

Figure 11: The experimental results of IMPRESS on clean image. For the *Style Mimicking* task, the "Clean Image" column presents results generated by a model using clean images as fine-tuning data, and the "Purified Clean Image" column shows results from models using clean images purified by IMPRESS as fine-tuning data. For the *Malicious Editing of Images* task, the "Clean Image" column represents the results of image editing using clean images as the target, while the "Purified Clean Image" column presents the results of image editing using clean images that have been purified by IMPRESS as the target.

Table 7: All Artists Results

| Artist | Metric (%) | Clean | Protected | Post-processed by IMPRESS |
|---|---|---|---|---|
| Raphael Kirchner | CLIP classifier Acc | 88.0 | 77.0 | 91.0 |
| | Diffusion classifier Acc | 93.0 | 87.0 | 93.0 |
| Camille Pissarro | CLIP classifier Acc | 80.0 | 2.0 | 88.0 |
| | Diffusion classifier Acc | 100.0 | 5.0 | 61.0 |
| Pyotr Konchalovsky | CLIP classifier Acc | 76.0 | 34.0 | 68.0 |
| | Diffusion classifier Acc | 90.0 | 41.0 | 90.0 |
| Childe Hassam | CLIP classifier Acc | 99.0 | 53.0 | 95.0 |
| | Diffusion classifier Acc | 98.0 | 47.0 | 65.0 |
| Paul Cezanne | CLIP classifier Acc | 93.0 | 6.0 | 73.0 |
| | Diffusion classifier Acc | 100.0 | 31.0 | 98.0 |
| Claude Monet | CLIP classifier Acc | 93.0 | 20.0 | 95.0 |
| | Diffusion classifier Acc | 100.0 | 16.0 | 91.0 |
| Albrecht Durer | CLIP classifier Acc | 99.0 | 78.0 | 91.0 |
| | Diffusion classifier Acc | 89.0 | 82.0 | 81 |
| Eugene Boudin | CLIP classifier Acc | 98.0 | 70.0 | 95.0 |
| | Diffusion classifier Acc | 93.0 | 34.0 | 80.0 |
| Average | CLIP classifier Acc | 90.8 | 42.5 | 87.0 |
| | Diffusion classifier Acc | 95.4 | 42.9 | 82.3 |

# I Comparison with More Baseline Methods

In this section, we compare our proposed IMPRESS with four additional baseline methods: 1) traditional denoising method (e.g., low-pass filtering); 2) enhanced preprocessing approaches (re-

size+flipping+JPEG compression); 3) robust training methods (e.g., Radiya-Dixit et al. (2021a)); 4) advanced denoising approaches (e.g., Diffpure (Nie et al., 2022))

Table 8: Comparison of IMPRESS with other baselines on Style Mimicking tasks

| Metric | Clean | Protected | Post-processed | | | |
| --- | --- | --- | --- | --- | --- | --- |
| | | | Low-pass Filtering | Resize + Flip + JPG | Robust Training | Ours |
| CLIP classifier Acc | 90.8 | 42.5 | 51.3 | 52.9 | 46.5 | **87.0** |
| Diffusion classifier Acc | 95.4 | 42.9 | 52.1 | 52.3 | 47.0 | **82.3** |

Table 9: Comparison of IMPRESS with other baselines on Malicious Editing tasks

| Metric | Protected | Post-processed | | | |
| --- | --- | --- | --- | --- | --- |
| | | Low-pass Filtering | Resize + Flip + JPG | Diffpure | Ours |
| SSIM | 0.41 | 0.43 | 0.45 | 0.48 | **0.49** |
| PSNR | 14.24 | 14.52 | 14.68 | 14.62 | **15.61** |
| VIF-p | 0.11 | 0.12 | 0.14 | 0.14 | **0.16** |

Table 8 and 9 show the experimental results for those additional baselines. For the Style Mimicking task, we have incorporated additional baselines, namely low-pass filtering, resize + flipping + JPEG compression, and the robust training method by Radiya-Dixit et al. (2021a). For the Malicious Editing task, we have added low-pass filtering, resize + flipping + JPEG compression, and Diffpure (Nie et al., 2022) as baselines. It can be observed that our approach still achieved the best results comparing with those additional baselines. Note that robust training method (Radiya-Dixit et al., 2021a) is not tested on malicious editing task since it does not involve model fine-tuning. And Diffpure (Nie et al., 2022) is not tested on style mimicking task since it requires a pre-trained non-latent diffusion model on the target distribution (which is one of the major limitation of this method) but it is hard to find one on artist data currently.

## J  Computational Complexities

The extra computations needed for our method is also not very significant. Under our experimental setup, the purification of a single image took an average of 141 seconds. In comparison, GLAZE took an average of 90 seconds to compute adversarial noise for an image, while Photoguard required more than 600 seconds on average to perform the computations. In fact, due to the large size of the stable diffusion model, fine-tuning the model actually takes much longer time than optimizing those images, thus we believe the computational cost will Acceptable.

