# OpenReview forum: "IMPRESS: Evaluating the Resilience of Imperceptible Perturbations Against Unauthorized Data Usage in Diffusion-Based  Generative AI"
_NeurIPS.cc/2023/Conference — NeurIPS 2023 poster_

### Official Review · Reviewer_EvE8 · 2023-07-07

**Soundness:** 3 good
**Presentation:** 3 good
**Contribution:** 3 good
**Rating:** 7
**Confidence:** 4

**Summary:**

This paper investigates the reliability of methods for imperceptibly perturbing images to degrade the performance of latent diffusion models for style mimicry or editing. The authors observe that imperceptible perturbations designed to hamper diffusion-based image generation systems often result in perceptible artifacts when the “protected” image is reconstructed with a diffusion model. Noting that unprotected images typically do not exhibit such behavior, the authors propose a “purification” method in which images are optimized with respect to both a consistency objective (such that diffusion reconstruction preserves the purified image) and a similarity objective (such that purification does not perceptibly alter the image). The proposed purification method effectively mitigates two recent imperceptible perturbation methods for image editing and style transfer, indicating that stronger techniques are needed to protect image data from unwanted diffusion-based manipulation.


**Strengths:**

The issue of protecting intellectual property such as images from unauthorized use by the operators of generative models is timely and significant.

The proposed purification approach — essentially an image adversarial attack using a simple encoder-decoder consistency objective, projected gradient descent, and Gaussian perturbation initialization — appears straightforward to implement, and is effective in both fine-tuned style mimicry and one-shot image editing scenarios.

The authors evaluate their purification method against adaptive perturbations optimized with knowledge of the purification objective and find their method still remains effective

The authors include reasonable protection methods (GLAZE, Photoguard) and purification baselines (JPEG compression, Gaussian noise, resizing) in their experiments.





**Weaknesses:**

Figures 1 and 4 are unnecessarily hard to read. For the style mimicking experiments, the labels for each column ("clean image", "protected image", "purified image") misleadingly indicate that the images below are actual clean/protected/purified images from the dataset when in fact they appear to be images generated by a model fine-tuned on either clean, protected, or purified images. For the editing experiment, it would be better to actually show the input clean/protected/purified image in addition to the edited versions. This might require a tweak to the figure format or perhaps separate figures, but I think that more legible figures would better communicate the authors' proposed approach.

To justify the choice of tasks on which the proposed method is evaluated, the authors should more clearly indicate that they evaluate against two recent imperceptible protection methods on the specific tasks for which these protections methods were proposed. Otherwise, the choice of task (e.g. background editing) could come off as arbitrary.

The size of the experiment datasets are not stated (e.g. for style mimicry, the number of images per artist is given, but not the number of artists). The total number of samples evaluated should be given for both experiments.

Adaptive protection results are shown only for style mimicry, not editing.

The authors do not appear to discuss the runtime / computational complexity of their purification method.

In apparent contradiction with the authors' response to the NeurIPS checklist, compute details appear to be missing from the paper and appendix.

For the experiments section, the image examples are currently more compelling than the metrics. For the image editing task, it is not clear from Table 2 how much "better" the purified images fare than protected images -- e.g., how significant is a PSNR value of 15 vs. 14? Is there a value below which an attempted edit can be considered a failure? Without context, it is hard to get a sense of the effectiveness of the proposed approach. The classifier metrics for the style mimicry experiment are clearer, but still do not take into account many aspects of generated image quality (e.g. adherence to the prompt beyond style). A human subjects study assessing the quality and "correctness" of images generated in each task for the clean/protected/purified conditions would be much more persuasive.

The evaluation tasks appear to involve image models that operate under different "control" paradigms -- fine-tuning and text-prompting in the case of style mimicry, and one-shot image-conditioning and text-prompting in the case of background editing. This difference seems significant in that in the former case, purification must mitigate the influence of protective perturbations on the training process while in the latter case, purification must only mitigate the influence of protective perturbations on a forward pass through the diffusion model. I think this distinction between experimental tasks should be emphasized because (a) it is not immediately clear in the paper, and (b) because it is possible this could be framed as a strength of the proposed method (i.e. operating in distinct tasks/settings).

**Questions:**

It is interesting that the simple consistency objective is sufficient to defeat protective perturbations, especially when optimized only with knowledge of the image encoder and decoder. Does this result imply that existing protection approaches primarily leverage the image encoder and decoder rather than the latent diffusion model itself? It would also be interesting to see whether there are other efficient ways to approximately optimize through the latent diffusion process.

(The following is more of a thought / comment than a question). In the adversarial example literature, methods have been proposed to tackle optimization with respect to competing objectives (e.g. perturbation effectiveness and imperceptibility). Often, one objective dominates the optimization even when loss values are scaled with well-chosen weights, leading to poor results and inaccurate estimates of robustness. The method of Bryniarski et al. [1] seems to be effective in allowing adaptive adversaries to better balance multiple constraints. Such an approach might lead to more effective balancing of the similarity and consistency objectives not just for the IMPRESS purification method, but for an adaptive adversary like the one proposed in section 5.4 of the paper. I think the simple projected gradient descent method and loss formulation the authors propose in this paper is fine -- it shows the effectiveness of the core idea without the need for extensive tweaks -- but it would also be interesting to see future work incorporate more recent techniques from adversarial machine learning into the protection and purification methods.

[1] Bryniarski, Oliver and Hingun, Nabeel and Pachuca, Pedro and Wang, Vincent and Carlini, Nicholas. "Evading Adversarial Example Detection Defenses with Orthogonal Projected Gradient Descent." ICLR 2022.

**Limitations:**

Yes.

---

> ### Author Rebuttal · Authors · 2023-08-10
>
> # Response to Reviewer EvE8
>
> Thanks for the constructive comments!
>
> **Q1**:  Figures 1/4 are hard to read
>
> **A1**:  We are sorry for the confusion. Figures 1 and 4 showcase images generated by a model that was fine-tuned on either clean, protected, or purified images. We've detailed this in the captions of each figure. Displaying the input clean/protected/purified image alongside the edited versions is indeed a great idea. We will refine our paper based on your suggestion.
>
> **Q2**: Clarify the source of the task
>
> **A2**: We thank the reviewer for the suggestion. We will clarify in the experimental section of our paper that the task selection is based on the existing protection methods.
>
> **Q3**:  Number of evaluation samples
>
> **A3**:  We are sorry for the confusion. For the Style Mimicking task, we selected 8 artists and randomly extracted 24 pieces for fine-tuning the diffusion model and 100 pieces for validation, totaling 800 evaluation samples. For the Malicious Editing task, we selected 80 photos for testing, which amounts to 80 evaluation samples.
>
> In response to Reviewer UShu's feedback, we have expanded the datasets for both the Style Mimicking and Malicious Editing tasks by 50%. For the Style Mimicking task, we've added artworks from 4 new artists, and for the Malicious Editing task, we've incorporated an additional 40 photos. Consequently, the total number of evaluation samples has increased to 1200 for the Style Mimicking task and 120 for the Malicious Editing task. The updated results are shown in Table 1,2 in the pdf. It can be observed that the results after data supplementation are largely consistent with the original results. Additionally, the sample size of our test set has already surpassed that in the original GLAZE paper (1,170 evaluation samples) and the original Photoguard paper (60 evaluation samples).
>
> **Q4**:  Adaptive protection results for Malicious editing.
>
> **A4**:  We apologize for any inconvenience caused. We added the results in Table 7 in the pdf . We found that the adaptive protection method brings degradation in protection performance when applied to the Malicious Editing task, and fails to resist the image sanitization by IMPRESS.
>
> **Q5**:  Discussion on Computational Complexity / time cost
>
> **A5**:  Please see **CA3** in our response to all reviewers.
>
>
> **Q6**: Significance of Metrics and Introduction of Human Subjects
>
> **A6**: We agree that assessing the significance of the metrics used in the Malicious Editing task is quite challenging. The current evaluation metrics basically follows what is done in PhotoGuard and we also think this is not perfect. Therefore, we presented numerous image results in the paper to visually illustrate the purification effects of our proposed method. We agree with you that the human subjects study may lead to more convincing results and we plan to add it later. Also upon public release of the paper, we will disclose all experimental codes and datasets used, including images purified by our method, to provide a clear demonstration of the effectiveness of our purification approach for the public.
>
> **Q7**:  Emphasizing the Distinctions of Our Method Among Different Paradigms
>
> **A7**:  Thank you for your suggestion! As you pointed out, the GLAZE method focuses on affecting the fine-tuning process of image generative models, while the Photoguard method targets the model's forward pass. We will highlight this in both the introduction and experimental sections of the paper.
>
> **Q8**: Current Protection Methods Utilize the Encoder-Decoder alone
>
> **A8**:  Both the GLAZE method and the encoder attack of Photoguard primarily exploit the encoder and decoder components of the latent space diffusion model, especially the encoder. However, Photoguard's diffusion attack backpropagates through the entire diffusion model. Note our consistency objective of the encoder-decoder is also sufficient to overcome Photoguard's diffusion attack, which suggests that the encoder-decoder part is probably the key to their attack. This is also reasonable since the diffusion part is essentially reconstructing the latent vector.
>
> We believe that direct attacks on the diffusion process with imperceptible noise additions are challenging, as the continuous noise additions during the diffusion noising process would overwrite the subtle perturbations.
>
> **Q9**: Suggestion to use O-PGD
>
> **A9**: Thank you for the suggestion. We follow your advice and further tested the approach from Bryniarski et al. [2] within the adaptive method on the Style Mimicking task. The experimental results are shown in Table 3 in the pdf.
> We found that the method of Bryniarski et al. did not bring significant improvement to the adaptive protection method. This might be due to the objective of our multi-loss not aligning perfectly with that of Bryniarski et al.. This suggests that the underlying principles of adversarial attacks on diffusion models might differ from those on classification models. This offers a fresh perspective on the issue. Thank you very much!

---

> > ### Comment · Reviewer_EvE8 · 2023-08-14
> > **Response to rebuttal**
> >
> > I thank the authors for their detailed comments, and for performing additional experiments to address the concerns raised in my review. I think the rebuttal makes a strong case that the proposed method (a) is more effective than simple editing baselines and (b) at worst, operates at around the same computational complexity as the protection methods it counteracts. I appreciate the additional experiments for adaptive adversaries, including O-PGD, and agree with the authors that the lack of success there may be due to the difficulty of balancing the objectives. I think that incorporating the results and clarifications from the rebuttal, in addition to the modifications promised by the authors, will greatly strengthen the paper.
> >
> > Overall, I think the fact that (increasingly popular) image protection methods can be defeated with a straightforward adversarial attack -- and that corresponding adaptive methods do not easily defeat this attack -- is an interesting and important result. I have updated my score in response to the rebuttal.

---

> > > ### Author Response · Authors · 2023-08-14
> > >
> > > We sincerely appreciate the reviewer's insights and are thankful for the increased score post-rebuttal. It truly motivates our ongoing research efforts!

---

### Official Review · Reviewer_DYEa · 2023-07-08

**Soundness:** 3 good
**Presentation:** 4 excellent
**Contribution:** 3 good
**Rating:** 6
**Confidence:** 5

**Summary:**

This paper proposes IMPRESS, a new purification method to remove imperceptible perturbations that are claimed to be able to prevent unauthorized data usage in diffusion-based image generation. Specifically, IMPRESS aims to produce a purified version of the perturbed image by minimizing the distance between the perturbed image and its reconstructed image as well as the distance between the perturbed image and the purified image. The authors also consider a challenging scenario where protection can be adapted to IMPRESS.

**Strengths:**

- Very good writing and thorough review of related work
- Clear motivation, especially for the elegant design of the consistency loss
- Sufficient descriptions of technical details of IMPRESS and the experimental settings
- Two SOTA protection methods in two tasks are considered.

**Weaknesses:**

- Limited comparison to (potential) baselines.
1) First of all, as acknowledged by the authors in Section 5.3, previous work (Salman et al. and Sandoval-Segura et al., 2023) has already identified the effectiveness of JPEG compression in solving the studied problem. Therefore, it should be explicitly stated in the Introduction that this paper is not the first to study whether “Imperceptible Perturbations Really Prevent Unauthorized Data Usage”. One specific question: What does it mean by “compression rate as 15%”? Is the quality factor 15% (stronger compression) or 85%? This is important because Sandoval-Segura et al. 2023 have found that choosing strong compression is the key to purifying the perturbations.
2) In addition, the countermeasures that are considered in the original papers of GLAZE and PhotoGuard should also be compared. This is necessary because this paper considers two different metrics (i.e., style classification scores) for measuring the effectiveness of IMPRESS. It is not clear whether those originally considered countermeasures can already achieve very good results in these two metrics.
3) Other popular purification methods should also be tested. For example, existing work has shown that diffusion-based purifications are very effective in removing adversarial perturbations. Such methods should be definitely compared.

- Limited explorations of adaptive attacks. Table 3 demonstrates that adaptive attacks show similar results and even worse results than non-adaptive attacks. This is surprising since additional knowledge is incorporated, and it also gives a sign that the adaptive attack may not be properly designed. The authors should either consider better designs or explain why this weird phenomenon happens.

Typos:

Line 232: should be some grammatical mistake for “setting reproduced the code”

Line 309: whether is possible -> whether it is possible

**Questions:**

See weaknesses

**Limitations:**

I do not find anywhere discussing Broader Impacts and Limitations, although the authors check "yes".

---

> ### Author Rebuttal · Authors · 2023-08-10
>
> # Response to Reviewer DYEa
>
> Thanks for the constructive comments!
>
> **Q1**: More Baselines
>
> **A1** Please see **CA2** in our common response to all reviewers
>
> **Q2**:  The first to study Imperceptible Perturbations
>
> **A2**:  Thank you for your suggestion. What we want to say here is prior works are only targeting the PhotoGuard method alone and there is no systematic study on the general type of imperceptible perturbations. Nonetheless, we admit that the claim in the paper may sound a bit misleading and we will revise the claims made in our paper.
>
> **Q3**:   Interpretation of "JPEG compression rate"
>
> **A3**:  We apologize for any confusion caused. In our paper, when we mention "compression rate of 15%", we are referring to a quality factor of 15%. This implies that the storage space occupied by the compressed image is only 15% of the original, representing a stronger compression. Additionally, we have appended experimental results under other JPEG compression quality factors for the Style Mimicking task, which is shown in Table 6 in the pdf. It's evident that the best performance is achieved when the quality factor is at 15%. We observed that a lower quality factor leads to significant image distortion.
>
> **Q4**:  . Inefficacy of the adaptive protective method:
>
> **A4**:   Please see **CA1** in our response to all reviewers.
>
> **Q5**:  Minor typos
>
> **A5**:  Sorry for the confusion. We have revised the paper as suggested.

---

> > ### Comment · Reviewer_DYEa · 2023-08-14
> > **Raised score**
> >
> > I thank the authors for providing the rebuttal, which contains a lot of new results that well address my concerns.

---

> > > ### Author Response · Authors · 2023-08-14
> > >
> > > We sincerely appreciate the reviewer's thoughtful reconsideration of our paper following the rebuttal. Thank you for recognizing the improvements and adjustments we made!

---

### Official Review · Reviewer_ULFY · 2023-07-09

**Soundness:** 2 fair
**Presentation:** 3 good
**Contribution:** 2 fair
**Rating:** 5
**Confidence:** 3

**Summary:**

This work is focused on testing the vulnerability of approaches that prevent unauthorized data usage in diffusion models. The paper shows that existing approaches for this problem are fragile, and leads to difference b/w original and diffusion reconstructed image. Then, based on the previous observation the paper proposes a method named IMPRESS to purify the modified images based on a consistency loss. The results show that the proposed approach can successfully purify the imperceptible modifications on two existing approaches, GLAZE and PhotoGuard. Lastly, the work shows that an adaptive variant of GLAZE, that takes into account the proposed consistency loss still can't defend against IMPRESS.

**Strengths:**

1. The writing quality of the paper is satisfactory. The main approach is easy to follow, and important preliminaries are clearly explained.
2. The results show that the proposed approach can surpass simple baselines. Both PhotoGuard, and GLAZE function as a form of adversarial attack, and thus should be vulnerable to similar transformations such as JPEG Compression, and geometric transformations. The work considers these as baselines and shows for GLAZE the proposed approach works better. For PhotoGuard, all baselines perform equivalently and are able to purify the protected image.

**Weaknesses:**

1. The overall contribution of this work may not be significant. The work relies on solving an expensive optimization problem and shows that imperceptible perturbation cannot prevent unauthorized data usage. However, it also shows that simple preprocessing approaches such as JPEG, and Resizing are effective against PhotoGuard. A similar observation was already made by [1]. The results on GLAZE show better performance, but the work doesn't consider more optimized baselines such as what happens for different JPEG compression ratios, or JPEG + Resizing + other geometric transforms.

2. The computational complexity of recovering the purified image is not discussed, especially w.r.t baselines.  How long does the proposed approach take? JPEG and Resizing are fairly inexpensive operations compared to the proposed approach that relies on solving an optimization problem.

[1] Sandoval-Segura, Pedro, Jonas Geiping, and Tom Goldstein. "JPEG Compressed Images Can Bypass Protections Against AI Editing." _arXiv preprint arXiv:2304.02234_ (2023).

**Questions:**

1. The approximation of consistency loss from the diffusion model to the encoder-decoder is unintuitive. Does this imply the IMPRESS approach is independent of the diffusion model?
2. Related to the above, how would the IMPRESS approach work for a non-latent diffusion model?

**Limitations:**

See the weaknesses above.

---

> ### Author Rebuttal · Authors · 2023-08-10
>
> # Response to Reviewer ULFY
>
> Thanks for the constructive comments!
>
> **Q1:** More Complicated Baselines
>
> **A1:** Please see **CA2** in our common response to all reviewers
>
> **Q2:** The Impact of Different JPEG Compression Rates
>
> **A2:** In our paper, we used the JPEG compression rate that provided the best protection. Table 6 in pdf shows the supplementary results for different JPEG compression rates. Please note that when we say "compression rate of 15%", it means we compress the image to only 15% of its original size, which is a rather strong compression.
>
> **Q3:** Computational Complexities
>
> **A3:** Please see **CA3** in our response to all reviewers.
>
>
> **Q4:** Is the IMPRESS approach independent of the diffusion model?
>
> **A4:** We only made the approximation for the sake of saving computational cost. The approximation is feasible since the diffusion layer’s goal is to **reconstruct the the original latent vector**.  Here's a detailed derivation:
>
> In a Latent Diffusion Model, denoted as $f_{LDM}$, the following steps are taken:
> $$
> z_0 = \mathcal{E}(x),
> \tilde{z_0} = diff(z_0),
> \tilde{x} = \mathcal{D}(\tilde{z_0})
> $$
> Here, $diff(.)$ represents the diffusion process. Initially, noise is added to $z_0$ over multiple steps, resulting in $z_1, z_2, .. z_T$. Next, the denoising model, $\epsilon_{\theta}(z_t, t, \tau_{\theta}(y))$, progressively denoises $z_T$ to obtain $\tilde{z}$. The model $\epsilon_{\theta}$ is trained using the following loss function:
> $$
> L_{LDM} = E_{\mathcal{E}(x), y, \epsilon \sim \mathcal{N}(0, 1), t }\Big[ \Vert \epsilon - \epsilon_{\theta}(z_{t},t, \tau_{\theta}(y)) \Vert_{2}^{2}\Big]
> $$
> As shown above, the predicted noise by the denoising model, $\epsilon_{\theta}(z_{t},t, \tau_{\theta}(y))$, aims to approximate the noise $\epsilon_t$ introduced during the forward process when $\tau_{\theta}(y) = 0$. Thus, in a well-trained LDM, the term $\Vert \epsilon_t - \epsilon_{\theta}(z_{t},t, \tau_{\theta}(y)) \Vert_{2}^{2}$ is nearly 0. This implies that $\tilde{z_0} \approx z_0$. Combining with the equation above, we deduce:
> $$
> f_{LDM} \approx \mathcal{D}(\mathcal{E}(x)), \text{when } \tau_{\theta}(y) = 0.
> $$
>
> IMPRESS can still be applied without the approximations (and potentially achieve even better results) at the cost of larger computational cost of backpropagation through the diffusion layers.
>
> **Q5:** Application of IMPRESS on Non-Latent Space Diffusion Models
>
> **A5:** As we mentioned in A4, IMPRESS can potentially be applied without the approximations (directly over the entire model), however, the current image protection techniques, (e.g., GLAZE and PhotoGuard), are primarily rely on latent space diffusion models. These techniques specifically "attack" the encoder-decoder components of these models. Hence, we cannot test the non-latent space diffusion models currently.

---

> > ### Comment · Reviewer_ULFY · 2023-08-14
> > **Raising Score**
> >
> > I thank the authors for the response and additional baselines. While the computational complexity of the proposed approach is high, compared to other baselines, the results show that the proposed approach works better (specifically for Style Mimicking). I've raised my score accordingly.

---

> > > ### Author Response · Authors · 2023-08-14
> > >
> > > Thank you for taking the time to re-evaluate our paper after the rebuttal. We're grateful for your constructive feedback and the positive shift in assessment!

---

### Official Review · Reviewer_vDR3 · 2023-07-10

**Soundness:** 3 good
**Presentation:** 3 good
**Contribution:** 3 good
**Rating:** 5
**Confidence:** 4

**Summary:**

In this work, the authors showed some intellectual protection methods for diffusion methods via imperceptible perturbations could be removed, hence questing the effectiveness of such methods. The authors observed perceptible inconsistency between the pair of original samples and the reconstructed ones by diffusion models, and then proposed IMPRESS method to disable these imperceptible perturbations. In the experiments, the authors tried to break two image protection methods, namely GLAZE and Photoguard, and showed their advantage over some known post-processing methods.

**Strengths:**

1). This work studies an interesting problem that questions the real effectiveness of the existing image protection methods.
2). The empirical observation between the differences of protected images and reconstructed images seem appealing.
3). The paper presents well and is easy to follow.


**Weaknesses:**

1). The two terms in the proposed loss (in Eq.4.1) are common denoising operators and are straightforward to come up with. This may hinder the technical novelty of this work.
2). There have been many denoising methods in the computer vision community, either deep learning based or conventional methods. Could these methods be modified as a proxy of the proposed method?
3). This paper lacks baseline methods. Why not compare with advanced denoising methods?
4). Though the methods to break are based on imperceptible perturbations, there can also be perceptible ones, please discuss whether your method can deal with such perturbations.


**Questions:**

See Weakness.

**Limitations:**

NA.

---

> ### Author Rebuttal · Authors · 2023-08-10
>
> # Response to Reviewer vDR3
>
> Thanks for the constructive comments!
>
> **Q1**: limited novelty
>
> **A1**: We respectively disagree and believe that our work is novel in the following aspects:
>
> 1) The issue we delve into (vulnerabilities of imperceptible perturbations based defenses) is new and critical (as also mentioned by Reviewer EvE8). Currently there is no rigorous work out there to pinpoint potential vulnerabilities within these protection methods.
>
> 2) The observations and perspectives we bring in this paper is new: existing image protection methods with imperceptible perturbations can result in noticeable differences between the protected image and the reconstructed image.
>
> 3) Similar loss terms does not necessarily mean the techniques are the same (e.g., most model training works share the same cross-entropy loss but it does not mean they are all the same). Removing noise for diffusion model’s training data is a new task that is totally different from prior ones.
>
> **Q2**: Employ computer vision denoising techniques & incorporate advanced baselines.
>
> **A2**: Please see **CA2** in our response to all reviewers.
>
> **Q3**: What about perceptible perturbations?
>
> **A3**: Our method is build upon the assumption that the perturbation noise is inpercpetible and the losses are designed based on such an assumption. This assumption is also reasonable as percpetible perturbations can be easily identified by the model trainer and thus have higher chance to be discarded during model fine-tuning. Currently, the protection with perceptible perturbations is beyond the scope of this paper. However, this is an intriguing idea, and we are inclined to delve deeper into this matter in our subsequent research.

---

> > ### Comment · Reviewer_vDR3 · 2023-08-15
> >
> > I thank the authors for preparing the responses which partially address my concerns. However, I still have some main concerns. E.g., First, for existing unauthorized data protection methods, only two methods, i.e., PhotoGuard and GLAZE were investigated, without any theoretical analysis, how can the authors ensure the proposed method/claims be still valid for other or future unknown protection methods? Besides, the imperceptible perturbations were mainly based on L_inf norm, what if adopting semantic manipulations (such perturbations can be imperceptible as well)? Thus, I would still maintain the score.

---

> > > ### Author Response · Authors · 2023-08-15
> > >
> > > Thank you for your feedback and overall positive attitude on our work! We are glad that part of your concerns are resolved and we are happy to further clarify on your remaining concerns.
> > >
> > > **RQ1**: Our method against unknown future protection techniques.
> > >
> > > **RA1**: First, we want to emphasize that our goal is not to break all future protection techniques. We develop this framework to find the potential vulnerabilities in **existing protection methods**, point out their failure cases and severity, hopefully further guide future protection researches. As mentioned by reviewer EvE8, we believe our research on this matter is both timely and significant.
> > >
> > > Nonetheless, we believe our IMPRESS method can potentially extend to other unseen imperceptible perturbation based defenses on LDMs as our design is quite general based on the structural characteristics of the latent diffusion models. Thus the consistency loss term in IMPRESS can still be effective for future defenses.
> > >
> > > **RQ2**: Didn’t consider imperceptible semantic manipulations.
> > >
> > > **RA2**: We are sorry for the confusion. In fact, we have considered the imperceptible semantic manipulations in our work: the GLAZE method employs constraints based on LPIPS score which is a metric that leverages semantic similarity and doesn't place special demands on the $\ell_{\infty}$ distance.  And results suggest that this LPIPS based semantic manipulation also can be purified by the IMPRESS method.
> > >
> > >
> > > Finally we want to thank you again for the constructive feedback during the review process. If there are any lingering ambiguities or concerns, please let us know. Many thanks!

---

> > > > ### Comment · Reviewer_vDR3 · 2023-08-18
> > > >
> > > > Thanks for the explanations.  Though I agree it is important to examine potential vulnerabilities, I don't think it is compelling enough by investigating two data protection methods only.

---

> > > > > ### Author Response · Authors · 2023-08-18
> > > > >
> > > > > We are glad that our explanations make it clearer to you! We understand your point. Though this line of research is still in the early stages currently with only two methods publicly available, we hope our work can set up a testbed for future protection works (which we believe will be increasingly popular soon) and help the community develop truly effective protection methods.

---

### Official Review · Reviewer_UShu · 2023-07-12

**Soundness:** 3 good
**Presentation:** 3 good
**Contribution:** 2 fair
**Rating:** 5
**Confidence:** 3

**Summary:**

Diffusion-based image generation models, such as Stable Diffusion or DALLE2, are able to learn from given images and generate high-quality samples following the guidance from prompts. However, such ability also brings unauthorized data usage problem. In response, several attempts have been made to protect the original images from this problem by adding imperceptible perturbations. The author discovered that it is possible to render these methods ineffective by purifying the image and disabling the added perturbation. They conduct experiments for style mimicking and malicious editing tasks to comprehensively evaluate the risks of using imperceptible perturbations.

**Strengths:**

1.	This paper identifies a common drawback in existing data protection methods by adding imperceptive perturbations.
2.	The method proposed by the authors is straightforward in its implementation but yields impressive results.


**Weaknesses:**

1.	The method proposed in this article is based on empirical findings and lacks profound theoretical support, which is slightly incongruent with the theme of the NeurIPS conference.
2.	This work has limited novelty because this approach is commonly used in the detection and removal of adversarial examples.
3.	The experimental results exhibited in the article have limited scope and quantity, which makes it difficult to provide strong support for the effectiveness of the method.


**Questions:**

1.	Why the adaptive image protection with consistency-based losses in section 5.4 does not really bring true protection? What is the underlying reason behind it?
2.	How effective is the proposed method attacking against other methods which protect the original images from style mimicking or malicious editing?
3.	How do you solve the optimization problem in Eq.(4.1)? Is it through deep learning methods or mathematical methods, please provide some more details about the algorithm.

---------------------------------
I read the rebuttal and the questions are answered. My main concern is the novelty of the methodology.

**Limitations:**

It is recommended that the authors explain the limitations of the proposed method, such as potential negative social impact (i.e., someone may do malicious editing of images by using your approach),  limited applicable scenarios and so on.

---

> ### Author Rebuttal · Authors · 2023-08-10
>
> # Response to Reviewer UShu
>
> Thank you for your valuable comments!
>
> **Q1:**:  Lacks theoretical support for NeurIPS
>
> **A1:** Indeed our paper does not provide much theoretical results and we plan to leave the theoretical study as our future work. Nonetheless, we believe it still represent significant contributions as it points out an major drawback for existing imperceptible perturbations based defenses and new perspectives on the design of new protection methods. Also we believe empirical-based works are common in NeurIPS and it is a bit unfair to reject a paper simply because it is empirical-based.
>
> **Q2:** Similar approach is used in detection/removal of adversarial examples
>
> **A2:** We respectively disagree and believe our work in novel in several aspects:
> 1) The issue we delve into (vulnerabilities of imperceptible perturbations based defenses) is new and critical (as also mentioned by Reviewer EvE8, vDR3). Currently there is no rigorous work out there to pinpoint potential vulnerabilities within these protection methods.
>
> 2) The observations and perspectives we bring in this paper is new: existing image protection methods with imperceptible perturbations can result in noticeable differences between the protected image and the reconstructed image.
>
> 3) Similar loss terms does not necessarily mean the techniques are the same (e.g., most model training works share the same cross-entropy loss but it does not mean they are all the same). Due to diffusion model fine-tuning, removing those perturbation in our work is fundamentally different from removing adversarial examples (adversarial examples considered purely evasion attack while our case is more like poisioning as the user will fine-tune the model with the perturbed data).
>
>
> **Q3:** Insufficient experimental data.
>
> **A3:** We have supplemented our experiments with additional data (see Table 1, 2 in the pdf). We augmented the datasets for both GLAZE and Photoguard by 50%.  We observe that the results after supplementing the data align closely with the original results. We will add these results to our paper after completing tests on the baseline and reproducibility tests. Additionally, the data quantity in our test set already surpasses that in the test sets of both the GLAZE and Photoguard papers.
>
> **Q4:** Inefficacy of the adaptive protective method
>
> **A4:** Please see **CA1** in our response to all reviewers.
>
> **Q5:** Other image protection techniques
>
> **A5:** We apologize for the confusion. Currently, there are no other image protection techniques specifically designed for diffusion models. The two image protection techniques we tested (GLAZE and Photoguard) both emerged in the past six months. We will emphasize this point in our paper. If new image protection methods emerge, we will promptly follow up and test them.
>
> **Q6:** How to solve the optimization problem in Eq.(4.1)
>
> **A6:** We employ deep learning techniques to optimize our objective function. We set the perturbation budget to $p=0.1$, the regularization coefficient to $\alpha=0.1$, and use the Adam optimizer with a learning rate of $10^{-2}$ for 3,000 iterations. More details on the experiments can be found in Appendix B.
>
> **Q7:**: Clarify the limitations of the methodology and the scenarios in which it can be applied.
>
> **A7:** : Thank you for your comments, In our paper, we will include the following note: "Please be aware that the proposed IMPRESS method might be misused, such as eliminating the protective noise from the current protected artworks. This indicates that the current imperceptible perturbations for image generation models is still imperfect. We urge researchers to investigate more robust image protection techniques and advise caution when relying on existing protection technologies until more secure methods are developed"

---

### Author Rebuttal · Authors · 2023-08-10

# Response to Common Question for All Reviewers
We thank all the reviewers for your valuable comments!

**We have incorporated all experiments suggested by the reviewers:**

1. We've included a broader range of baseline methods, encompassing more potent traditional denoising techniques, combinations of various denoising methods, robust training strategies, and state-of-the-art denoising approaches. The results indicate that our method still outperforms others.
2. We've tested our approach on a more extensive dataset, and the results largely align with those presented in the paper.
3. We've augmented our experiments concerning adaptive defense techniques, which now include ablation studies on hyperparameters and stronger adaptive defense methods.
4. We've illustrated the purification effects across different JPEG compression quality factors.
5. We've added results for adaptive defense against Malicious Editing.

Besides, we have addressed all queries from the reviewers.

**Here we address some common questions for all reviewers:**

**CQ1:** Inefficacy of the adaptive protective method

**CA1:** We believe the suboptimal performance of the adaptive method is due to the following reasons:
1) The introduced supplementary loss term complicates the optimization of the loss function. For GLAZE method, the loss function of the adaptive method has three components. Optimizing three different loss components is typically challenging, and striking a balance among these components remains an intricate task.
2) The loss term employed for the adaptive method might clash with the inherent loss term of image protection techniques. The purpose of image protection technologies is to induce semantical differences between generated images and original ones. Our image purification approach aims to make the image maintains its semantic properties (consistency after reconstruction), potentially leading to underlying optimization conflicts.

To verify these issues, we've conducted two additional experiments:
1) We carefully tune the weight of the adaptive loss term. which is shown in Table 8 in the pdf.
We've adjusted the weight of the adaptive loss term over a broad spectrum, making it constitute approximately 10% to 95% of the total loss. We notice that when the proportion of the adaptive loss term is too minuscule within the entire loss, the experimental outcomes tend to resemble standard image protection technique results. Conversely, it considerably deteriorates the image protection performance. This suggests that achieving the balance of those three objective terms are hard.
2) As recommended by Reviewer EvE8, we further adopt Bryniarski et al.[1]'s Orthogonal Projected Gradient Descent approach to better balance competing objectives. The new results on GLAZE are shown in Table 3 in the pdf. We can observe that using O-PGD does not help much on the adaptive method.

These results suggest that designing adaptive image protection techniques specifically for IMPRESS might inherently be challenging.

**CQ2:** More baselines (e.g., CV denoising and robust training methods)

**CA2:** We thank for all the suggestions and added additional 4 types of baselines:
1) traditional denoising method (e.g., low-pass filtering); 2) enhanced preprocessing approaches (resize+flipping+JPEG compression); 3) robust training methods (e.g., [2]); 4) advanced denoising approaches (e.g., Diffpure [5])
Table 4 and 5 show the experimental results for those additional baselines. For the Style Mimicking task, we have incorporated additional baselines, namely low-pass filtering, resize + flipping + JPEG compression, and the robust training method by Radiya-Dixit, E., et al. [2]. For the Malicious Editing task, we have added low-pass filtering, resize + flipping + JPEG compression, and Diffpure[5] as baselines. It can be observed that our approach still achieved the best results comparing with those additional baselines. Note that robust training method [2] is not tested on malicious editing task since it does not involve model fine-tuning. And Diffpure [5] is not tested on style mimicking task since it requires a pre-trained non-latent diffusion model on the target distribution (which is one of the major limitation of this method) but it is hard to find one on artist data currently.

**CQ3:** Computational Complexities

**CA3:**  Indeed JPEG and resizing operations take neglectable time to compute, however, as our results suggest, they are far more ineffective compared with our proposed method. Meanwhile, we notice the extra computations needed for our method is also not very significant. Under our experimental setup (stated in Appendix B of the paper), the purification of a single image took an average of 141 seconds. In comparison, GLAZE took an average of 90 seconds to compute adversarial noise for an image, while Photoguard required more than 600 seconds on average to perform the computations. In fact, due to the large size of the stable diffusion model, fine-tuning the model actually takes much longer time than optimizing those images, thus we believe the computational cost will not be a big issue here.

We kindly request that you inform us of any remaining ambiguities or concerns. We are more than willing to address additional questions and conduct further experiments should the reviewers deem it necessary.


[1] Gowal, S., et al. Uncovering the limits of adversarial training against norm-bounded adversarial examples. arXiv preprint arXiv:2010.03593, 2020.

[2] Radiya-dixit, E., et al. Data poisoning won’t save you from facial recognition. arXiv:2106.14851 (2021).

[3] Gowal, S.,  et al. Improving robustness using generated data. Advances in Neural Information Processing Systems, 34, 2021.

[4] Rebuffi, S.-A.,  et al. Fixing data augmentation to improve adversarial robustness. arXiv preprint arXiv:2103.01946, 2021.

[5] Nie, W, et al. "Diffusion models for adversarial purification." arXiv preprint arXiv:2205.07460 (2022).

---

### Decision · Program_Chairs · 2023-09-21

**Decision:**

Accept (poster)

**Comment:**

This paper scrutinized existing data protection methods, casting doubt on their actual effectiveness. The observation made is that these existing methods exhibit a high degree of fragility, with some simple and direct approaches proving capable of successfully breaching them. It garnered feedback from five reviewers, all of whom commended the intriguing observation and findings. The rebuttal included many new experiments and effectively addressed most concerns.

The primary remaining concern centers around the technical simplicity of the proposed solution. The Area Chair acknowledges this concern but emphasizes that the key observation is exceptionally inspiring and offers valuable insights to the research community. Considering all factors, the Area Chair believes that the paper's overall strength outweighs its weaknesses and, as such, recommends acceptance. The authors should incorporate all the results and commitments from the rebuttal into the final version.